# DCBM: Data-Efficient Visual Concept Bottleneck Models

**Katharina Prasse** [* 1]  **Patrick Knab** [* 2]  **Sascha Marton** [2]  **Christian Bartelt** [2]  **Margret Keuper** [1 3]

## Abstract

Concept Bottleneck Models (CBMs) enhance the interpretability of neural networks by basing predictions on human-understandable concepts. However, current CBMs typically rely on concept sets extracted from large language models or extensive image corpora, limiting their effectiveness in data-sparse scenarios. We propose Data-efficient CBMs (DCBMs), which reduce the need for large sample sizes during concept generation while preserving interpretability. DCBMs define concepts as image regions detected by segmentation or detection foundation models, allowing each image to generate multiple concepts across different granularities. Exclusively containing dataset-specific concepts, DCBMs are well suited for fine-grained classification and out-of-distribution tasks. Attribution analysis using Grad-CAM demonstrates that DCBMs deliver visual concepts that can be localized in test images. By leveraging dataset-specific concepts instead of predefined or general ones, DCBMs enhance adaptability to new domains. The code is available at: https://github.com/KathPra/DCBM.

## 1. Introduction

Deep neural networks (DNNs) achieve state-of-the-art performance across various domains, yet their opaque nature limits trust, particularly in safety-critical applications. To address this, Explainable Artificial Intelligence (XAI) strives to make model decisions more transparent and interpretable. Concept Bottleneck Models (CBMs) (Koh et al., 2020) are an inherently interpretable framework, as they base predic-

*Equal contribution  [1]Data and Web Science Group, University of Mannheim, Mannheim, Germany  [2]Clausthal University of Technology, Clausthal, Germany  [3]Max-Planck-Institute for Informatics, Saarland Informatics Campus. Correspondence to: Katharina Prasse <katharina.prasse@uni-mannheim.de>, Patrick Knab <patrick.knab@tu-clausthal.de>.

*Proceedings of the 42nd International Conference on Machine Learning*, Vancouver, Canada. PMLR 267, 2025. Copyright 2025 by the author(s).

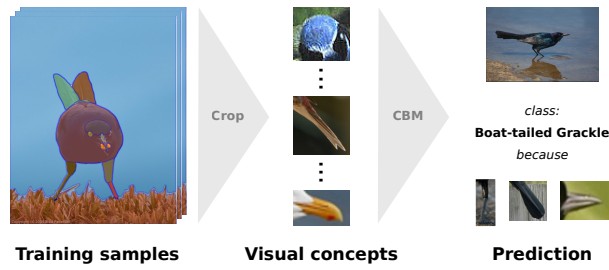

*Figure 1.* **DCBMs extract image regions as concepts.** Using vision foundation models, we use crop image regions as concepts for CBM training. Based on few concept samples (50 imgs / class), DCBMs offer interpretability even for fine-grained classification.

tions on a weighted combination of human-understandable concepts. While early CBMs employ manually crafted concepts (Koh et al., 2020), recent advances in vision-language models have enabled text-aligned CBMs, which leverage textual descriptions for concept selection and interpretability (Menon & Vondrick, 2023; Oikarinen et al., 2023; Yang et al., 2023). Given that precise class descriptions are available, text alignment can effectively guide the concept selection towards representative semantic concepts. However, recent works highlight that CBMs benefit from concepts extracted in the visual domain (Kowal et al., 2024; Rao et al., 2024; Sun et al., 2024; Wang et al., 2023; Zhang et al., 2024a). This also increases their faithfulness against the reported image-text misalignment in the CLIP embedding space (Liang et al., 2022; Roth et al., 2023). Learning concepts (Rao et al., 2024) from large-scale datasets performs well for ImageNet classification, but is less suitable for fine-grained classification, where concepts must be defined at a specific granularity. Ideally, concepts can be recognized under domain shifts in order to increase their applicability in real-life use cases.

We propose **Data-efficient Visual CBMs (DCBM)** to improve interpretability in data-scarce domains. DCBMs use segmentation and detection foundation models to extract image regions as visual concept proposals. This approach generates multiple concepts at different levels of granularity from a single image, allowing to create a meaningful concept bank even in data-scarce settings. Our design en-

ables flexible adaptation to novel datasets while exclusively containing dataset-specific concepts. For wide applicability, DCBMs leverage the popular CLIP embedding space e.g. (Rao et al., 2024; Yang et al., 2023). By extracting concepts in the visual domain, we avoid bridging the gap between the modalities (Liang et al., 2022), while being able to name our generated concepts using the image-text alignment. DCBMs enable fine-grained concept differentiation and maintain interpretability in out-of-distribution (OOD) settings.

Our main contributions can be summarized as follows:

- With DCBMs, we introduce a simple and data-efficient framework that can easily be adapted to novel datasets.

- We create concepts using segmentation and detection models which are inherently dataset-specific and offer multiple concept granularities. We generate concepts in the visual domain to avoid modality gaps and to be independent of textual descriptions.

- We extensively evaluate the DCBM framework across concept proposal generation methods and datasets. To ensure its usefulness for real-life applications, we assess the transferability of DCBMs to out-of-domain (OOD) settings and verify the localization of the important concepts.

## 2. Related work

Explainable artificial intelligence offers numerous approaches to making model decisions interpretable. Post hoc methods create an explanation on top of an existing model based on the given model prediction. By leveraging activations (Selvaraju et al., 2019), the relevant image area is highlighted or model embeddings (Yuksekgonul et al., 2023) are used to show concept activations. Their interpretability is limited, as one cannot be sure that their explanations are inherent to the model. Ante hoc methods, however, are inherently interpretable and provide predictions along with explanations, e.g. (Chefer et al., 2021; Menon & Vondrick, 2023; Oikarinen et al., 2023; Rao et al., 2024; Yang et al., 2023). They include single-layer linear neural networks (CBMs), e.g. (Koh et al., 2020), decision trees, e.g. (Mahbooba et al., 2021), and self-explaining networks, e.g. DEAL (Li et al., 2025), BCOS-networks (Böhle et al., 2022), and VLMs such as LLaVA-NeXT (Li et al., 2024b). While other data-efficient methods exist, e.g., prompt-tuned CLIP, we focus exclusively on interpretable models.

### 2.1. Concept bottleneck models

This family of models predicts image classes based on weighted linear combinations of concepts (Koh et al., 2020). CBMs mainly differ in the way they extract concepts.

Menon & Vondrick (2023) generate GPT-3 descriptions of class-specific concepts, providing insights into the visual attributes associated with each category (DCLIP). Similarly, Oikarinen et al. (2023) generate concept sets for CBMs using GPT-3 for training their label-free CBM (LF-CBM). Panousis et al. (2023) extend LF-CBM by incorporating a concept discovery module (CDM) that encourages sparsity by learning which subset of concepts is relevant for a given image. Language-guided CBM (LaBo) employs GPT-3 for concept creation, but it stands out by using a submodular function to select concepts from candidate sets with the goal of retaining the most discriminative and least overlapping concepts (Yang et al., 2023). All aforementioned methods incorporate text in their concept discovery process. We argue that multi-modality concept discovery adds nuances to image classification interpretability, which are not inherently there, in pure image classification models. Given that images and texts occupy distinct areas of the CLIP embedding space (Liang et al., 2022), we argue that only image-level concepts allow for true interpretability. Following the same line of argumentation, Rao et al. (2024) introduce Discover-then-Name (DN-CBM), which employs sparse autoencoders for image-level concept generation. These concepts are mapped to names derived from a corpus of broad textual descriptions, and the resulting mapped concepts are used to train a CBM. Parallel work by Schrodi et al. (2024) understands concepts as low-rank approximations of model activations, which they learn using non-negative matrix factorization on image embeddings. Moreover, self-supervision can be used to learn concepts from data (Alvarez Melis & Jaakkola, 2018; Wang et al., 2023). Following the same motivation, we and parallel works have employed foundation models to propose visual concepts from data (Kowal et al., 2024; Sun et al., 2024; Zhu et al., 2024). We argue that concepts should have high granularity and be dataset-specific to allow adaptation to varying target domains. We use CBMs in their original form - a single linear layer - while there exist variants learning residuals (Zabounidis et al., 2023), investigating locality (Raman et al., 2023), and assessing concept correlations (Heidemann et al., 2023).

### 2.2. Visual concept generation

CBMs typically contains autoencoders (Huben et al., 2024; Rao et al., 2024), non-negative matrix factorization (Fel et al., 2023; Schrodi et al., 2024; Zhang et al., 2021), K-Means (Ghorbani et al., 2019), or PCA (Zhang et al., 2021; Zou et al., 2023). Panousis et al. (2024) share our conviction that multi-level concepts enhance the expressiveness of CBMs. In contrast to their 2-level approach, our DCBM can detect multi-level concepts. Following other XAI methods, we generate concept proposals in an unsupervised manner using segmentation foundation models (Kowal et al., 2024;

Sun et al., 2024; Zhu et al., 2024). Specifically, we extract image crops using segmentation (Carion et al., 2020; He et al., 2017; Kirillov et al., 2023; Li et al., 2024a; Ravi et al., 2025) and detection foundation models (Liu et al., 2024), with hyperparameters tuned to capture both part-level and instance-level concepts. We can steer promptable detection methods more than generic models and expect to return fewer and more class-relevant concepts. To minimize redundancy, we cluster the generated concept proposals. In contrast to BotCL (Wang et al., 2023), we create visual concepts instead of learning them. We evaluate the correct localization of concepts using the Grid Pointing Game (Bohle et al., 2021). In comparing concept activations of the corresponding class to activations of three random other classes, we evaluate whether the concepts are class-specific.

## 2.3. Image-level interpretability

Besides CBMs, part-based classifiers (Chen et al., 2019; Donnelly et al., 2022; Nauta et al., 2021; Pham et al., 2024; Wang et al., 2021; Xue et al., 2024) and attribution methods (Bohle et al., 2021; Knab et al., 2025; Ribeiro et al., 2016; Selvaraju et al., 2019; Zhang et al., 2018) offer interpretability in the visual domain. In contrast to CBMs, part-based classifiers are often trained end-to-end (Chen et al., 2019). While their prototypes have visual similarities with DCBM concepts and their explanations share similarities with CBMs, the fact that prototypes are learned in a supervised manner, e.g. (Chen et al., 2019; Pham et al., 2024), sets them apart. BotCL (Wang et al., 2023) learns concepts in a self-supervised manner and requires the number of concepts to learn as a hyperparameter. Given that prototypes are generated during training time, attribution methods can be used to highlight relevant image areas during inference (Chen et al., 2019).

Attribution methods trace model decisions back to the image plane, e.g. (Bohle et al., 2021; Knab et al., 2025; Ribeiro et al., 2016; Selvaraju et al., 2019). They determine the relevant image region by using, e.g., gradients, activations, occlusions, or perturbations. Grad-CAM (Selvaraju et al., 2019), for instance, visualizes the activations of the network's final layer, while Local Interpretable Model-agnostic Explanations (LIME) (Knab et al., 2025; Ribeiro et al., 2016) uses perturbation-based surrogates to approximate interpretable models. For DCBM, we evaluate the localization of concepts using the Grid Pointing Game to validate the concepts in the test images, following the same intuition as Wang et al. (2023).

## 3. Data-efficient concepts for CBMs (DCBM)

The design of DCBM is driven by the focus on data-efficiency, simplicity, and step-wise inspectability. To this end, we define concepts as image regions that we generate

using segmentation or detection foundation models. The initial concept set is clustered to reduce redundancies. In line with the main idea of CBMs, we select simple methods (k-means) to achieve maximum interpretability. DCBMs can be inspected at every step, (1) foundation model concept proposals, (2) cluster centroids, and (3) CBM explanations. On top of that, attribution methods visualize inference-time concept activations to provide interpretations grounded in the image plane. Figure 2 illustrates the framework that derives visual concepts, which are subsequently named using the CLIP space.

### 3.1. Concept proposal generation

We use foundation models for segmentation and detection to facilitate data-efficient concept generation and extract multiple concepts from the same image. Our choices are the foundation models Segment Anything 1 & 2 (Kirillov et al., 2023; Ravi et al., 2025), GroundingDino (Liu et al., 2024), MaskRCNN (He et al., 2017), and DETR (Carion et al., 2020). We compare the segmentation models, which require no input, to the open-set detection method GroundingDino, in which we use various concept sets from the literature as input. We construct a concept proposal subset by selecting as few as 50 random training images per class. This has the goal of mitigating the risk of overfitting while increasing efficiency since not all training images are used for concept generation. Drastically reducing the number of images needed for concept generation makes DCBM highly data-efficient, i.e., for ImageNet, DCBM uses (-96%) of training images for concept generation compared to other data-driven CBMs. Further discussions on data-efficiency can be found in Appendix A.1.

Selecting 50 images is a hyperparameter, for CUB, which contains fewer samples, we include all available images in the concept generation set $\mathcal{D}$. Especially for large datasets such as ImageNet and Places365, this reduces the number of images considered during concept extraction by 96% and 99%, respectively, compared to other methods (Schrodi et al., 2024). Task-agnostic pre-training (Rao et al., 2024) requires large-scale datasets such as CC3M (Sharma et al., 2018). Efficiency can be increased by using fewer training samples at the cost of slight performance degradations, as shown in Appendix E.1. After receiving the foundation model segmentation or detection results, we crop the images in $\mathcal{D}$ according to the bounding box and add the resulting sub-image to the concept proposal set $\mathcal{S}$. For each image in $\mathcal{D}$, multiple sub-images can be created, thus $|\mathcal{D}| \leq |\mathcal{S}|$. The removal of background information is ablated in Appendix D.1. To refine the concept proposals, concepts that are below or above a given size threshold can be excluded from $\mathcal{S}$, as evaluated in Appendix E.2.

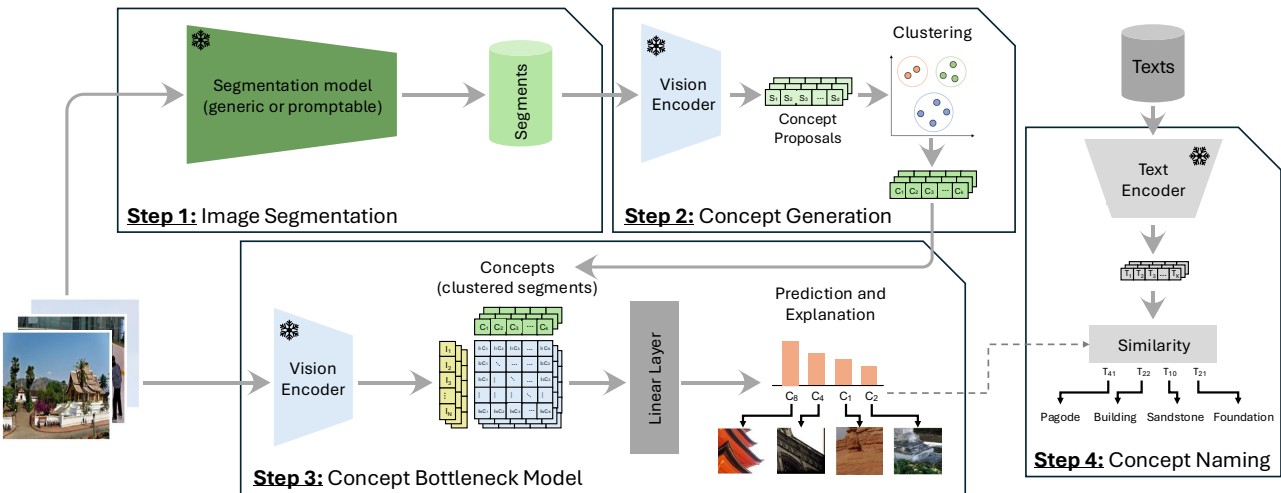

*Figure 2.* **DCBM framework.** The DCBM framework generates concept proposals through foundation models (Step 1). These proposals are then clustered, each represented by its centroid (Step 2). Finally, the unique concepts are utilized to train a sparse CBM, effectively (Step 3). We leverage the image-text alignment to map the visual concept to the corresponding textual concept (Step 4). We can remove undesired concepts after Step 2.

## 3.2. Concept generation

We embed the concept proposal set $\mathcal{S}$ using a frozen image encoder with backbone architecture $f$ such that $\mathcal{S}_{\text{enc}}^d = f(\mathcal{S}) = (f(\mathcal{S}_i))_{i=1}^d \in \mathbb{R}^{d \times m}$, where $m$ denotes the embedding space's dimensionality and $d$ the number of concept proposals in $\mathcal{S}$. To reduce redundancy while maintaining high variance of concepts, we cluster the concept proposal embeddings $\mathcal{S}_{\text{enc}}^d$. In Appendix D.3, we ablate the choice of clustering algorithm by comparing centroid-based K-Means with agglomerative clustering. Additionally, in Appendix D.5, we analyze the impact of varying the number of clusters in K-Means. We select a rather large number of clusters, since we aim to create concise concepts. For each cluster, we define its centroid $c_j$ as a concept, thus obtaining $\mathcal{C} = \{c_j\}_{j=1}^k \in \mathbb{R}^{k \times m}$ in the embedding space. We ablate two approaches for determining the centroid $c_j$ of cluster $j$: the mean and the median of the clustered embeddings, as reported in Appendix D.4. The concept set $\mathcal{C}$ constitutes the final concept set employed for training the CBM model.

**Concept Removal.** DCBM allows a targeted removal of specific, undesired concepts after the concept generation phase (Step 2 in Figure 2). Leveraging CLIP's multimodal capabilities, we specify the concept to be removed using a textual prompt. For example, to exclude the concept "*stone*", we compute its text embedding and remove all visual concepts with cosine similarity above the threshold. This capability allows for fine-grained control to exclude concepts, allowing users to suppress spurious correlations or explicitly highlight desired causal factors (see Appendix G.2 for further details).

## 3.3. Concept bottleneck model

The CBM learns a linear mapping function $h(\cdot)$ that transforms the concept activations into corresponding class label predictions such that

$$t(\mathbf{x}_i) = h(a(\mathbf{x}_i)) = \omega^\top a(\mathbf{x}_i), \qquad (1)$$

where $t(\mathbf{x}_i)$ denotes the predicted label for input $\mathbf{x}_i$, $a(\mathbf{x}_i) \in \mathbb{R}^k$ represents the concepts associated with $\mathbf{x}_i$, and $\omega \in \mathbb{R}^k$ are the parameters of the linear model. To construct $a(\mathbf{x}_i)$, we compute the projection of the embedding $f(\mathbf{x}_i)$ onto each cluster centroid $c_j$, normalized by its squared $L_2$-norm as done by Yuksekgonul et al. (2023).

$$a(\mathbf{x}_i) = \frac{\langle f(\mathbf{x}_i), c_j \rangle}{\|c_j\|_2^2}, \quad \text{for } j = 1, \dots, k \qquad (2)$$

We train the model using the cross-entropy loss, in line with previous work (Rao et al., 2024). Additionally, we apply an $L_1$ regularization term on the parameters $\omega$ to promote sparsity in the learned weights,

$$\mathcal{L}_{\text{train}}(\mathbf{x}_i) = \text{CE}(t(\mathbf{x}_i), y_i) + \lambda \|\omega\|_1, \qquad (3)$$

where CE represents the cross-entropy loss, $\lambda$ is the sparsity hyperparameter that controls the influence of the regularization term, and $\omega$ are the parameters of the linear mapping.

## 3.4. Concept naming

An additional feature of DCBM is the naming of the concept set $\mathcal{C}$. Given that the backbone architecture $f$ is text-image aligned, we can embed a corpus of texts $\mathcal{T}$ into the

same embedding space as the images. Specifically, we define $\mathcal{T}_{\text{enc}}^d = f_{\text{text}}(\mathcal{T}) = (f_{\text{text}}(\mathcal{T}_i))_{i=1}^l \in \mathbb{R}^{l \times m}$, where $l$ represents the number of textual concepts within $\mathcal{T}$. It is important to note that $l$ may differ from $d$, because we utilize an open text corpus for this task. Each visual concept is assigned to the nearest textual feature based on cosine similarity. In line with previous work (Rao et al., 2024), we select $\mathcal{T}$ as common English words (Kaufman, 2012). The DCBM framework is further detailed in Algorithm 1 in Appendix A.

## 4. Evaluation

DCBMs exclusively incorporate dataset-intrinsic visual concepts, with additional benefits in data-efficiency and applicability to fine-grained classification tasks. We first outline the experimental setup (Section 4.1), followed by a quantitative performance assessment (Section 4.2), and a qualitative assessment of DCBM-generated explanations in Section 4.3. Lastly, we evaluate concept quality based on localization within the test images using Grid Pointing Game (Bohle et al., 2021) in Section 4.4.

### 4.1. Experimental setup

**Backbones.** The evaluation is conducted in the CLIP (Radford et al., 2021) embedding space with ResNet-50, ViT B/16, and ViT L/14 backbones. We compare different concept proposal methods: generic segmentation models, i.e., SAM2 (Ravi et al., 2025), SAM (Kirillov et al., 2023), DETR (Carion et al., 2020), Mask-RCNN (He et al., 2017), and Semantic-SAM (Li et al., 2024a), which segment the entire image, including background objects, and return concept proposals of the whole scene. Additionally, we employ a promptable detection model, i.e., GroundingDINO (Liu et al., 2024), in which we steer the concept generation toward relevant image regions. To this end, we use prominent part labels from the literature as detection prompts. This includes the CUB labels for bird parts (Wah et al., 2011), the sun attributes (Patterson et al., 2014), the animal attribute labels employed in AWA (Lampert et al., 2009), part-Imagenet labels (He et al., 2022), and Pascal parts labels (Chen et al., 2014). We ablate the prompts in combination with all datasets Appendix D.

**Datasets.** We evaluate DCBMs on the five commonly used datasets in the CBM community. For general image classification, we employ CIFAR-10, CIFAR-100 (Krizhevsky et al., 2009), and ImageNet (Deng et al., 2009), as they offer a wide range of classes. For domain-specific tasks, we use CUB (Wah et al., 2011) and Places365 (Zhou et al., 2017), which provide targeted, domain-specific categories. Additionally, we ablate DCBMs on ImageNette (Howard, 2019a) and the fine-grained dog classification dataset ImageWoof (Howard, 2019b). Moreover, we evaluate on awa2

(Xian et al., 2018), CelebA (Liu et al., 2015), and a subset of ImageNet (i.e. first 200 classes) to compare against other CBMs and to exemplify its applicability to diverse datasets. We further evaluate it on two novel datasets for the XAI community, the social-media dataset ClimateTV (Prasse et al., 2023) and MiT-States (Isola et al., 2015), as inspired by (Yun et al., 2023). We also evaluate on ImageNet-R (Hendrycks et al., 2021) to assess CBM performance under complete domain shifts. See Appendix B for a detailed dataset overview.

**Hyperparameters.** We ablate all hyperparameters on a held-out validation set. In case this does not exist, we construct it to comprise 10% randomly-selected, class-balanced training samples. Using this setting, we ablate on the CUB, ImageNette and ImageWoof datasets to find suitable hyperparameters. The search space comprises a learning rate $l_r = \{1e^{-4}, 1e^{-3}, 1e^{-2}\}$, sparsity-parameter $\lambda = \{1e^{-4}, 1e^{-3}, 1e^{-2}\}$ number of clusters $k = \{128, 256, 512, 1024, 2048, 4096\}$, and concept proposal models. For all datasets and concept proposal generation methods, $l_r = 1e^{-4}$ and sparsity parameter $\lambda = 1e^{-4}$ were found to be optimal. The number of clusters $k$ depends on the size of the concept proposal set $\mathcal{S}$, but we observed a tendency regarding greater values of $k$, and stick to $k = 2048$ if not mentioned otherwise. We train each DCBM model for 200 epochs with a batch size of 512 (Places365 & ImageNet) or 32 (all remaining). Concept clustering and CBM training takes five minutes for small datasets and up to two hours for large datasets on a single RTX A6000 (more in Appendix E.1).

### 4.2. DCBM performance

We benchmark DCBM on standard CBM datasets and assess the generalization capabilities of its visual concepts on the out-of-distribution dataset ImageNet-R. Moreover, we evaluate it on two uncommon datasets for XAI, MiT-States and ClimateTV, and the less common awa2 and CelebA datasets. Performance differences between datasets are marginal, exemplifying that DCBMs can be adapted to novel datasets.

#### 4.2.1. COMPARISON ON BENCHMARK DATASETS

In Table 1, we report a quantitative evaluation of DCBM's performance (top-1 accuracy) compared to recent CBM approaches, all of which leverage the text-image aligned backbones of CLIP. For each experiment, we include linear probe and zero-shot accuracy, with an asterisk (*) indicating performances reported in prior literature, and a dash (-) indicating no reported accuracies. As a standard, we choose 2048 concepts; ablations have shown that 4096 clusters improve ImageNet accuracy while CIFAR-10 accuracy increases with fewer clusters (256). This indicates that the number of concepts and the number of classes are related.

*Table 1.* **CBM benchmark.** Top-1 accuracy comparison across CBM models (CLIP ViT L/14). The highest accuracies are bolt. DNCB excels in fine-grained classification, while text-aligned and large-scale vision CBMs prevail on general datasets.

| Model | CLIP ViT L/14 | | | | |
|---|---|---|---|---|---|
| | IMN | Places | CUB | Cif10 | Cif100 |
| Linear Probe ↑ | 83.9* | 55.4 | 85.7 | 98.0* | 87.5* |
| Zero-Shot ↑ | 75.3* | 40.0 | 62.2 | 96.2* | 77.9* |
| LF-CBM (Oikarinen et al., 2023) ↑ | - | 49.4 | 80.1 | 97.2 | 83.9 |
| LaBo (Yang et al., 2023) ↑ | **84.0*** | - | - | 97.8* | 86.0* |
| CDM (Panousis et al., 2023) ↑ | 83.4* | 55.2* | - | 95.9 | 82.2 |
| DCLIP (Menon & Vondrick, 2023) ↑ | 75.0* | 40.5* | 63.5* | - | - |
| DN-CBM (Rao et al., 2024) ↑ | 83.6* | **55.6*** | - | **98.1*** | **86.0*** |
| DCBM-SAM2 (Ours) ↑ | 77.9 | 52.1 | **81.8** | 97.7 | 85.4 |
| DCBM-GDINO (Ours) ↑ | 77.4 | 52.2 | **81.3** | 97.5 | 85.3 |
| DCBM-MASK-RCNN (Ours) ↑ | 77.8 | 52.1 | **82.4** | 97.7 | 85.6 |

*Table 2.* **Ood performance.** Error rate changes compared between visual CBMs (CLIP ViT-L/14) on ImageNet-R classes (and a baseline). DCBM loses less performance when moving from iid to ood evaluations, compared to the task-agnostic DN-CBM.

| | IN-200 | IN-R | Gap(%) |
|---|---|---|---|
| DN-CBM (Rao et al., 2024) ↓ | 16.4 | 55.2 | 38.8 |
| DCBM-SAM2 (Ours) ↓ | 21.1 | **48.5** | **27.4** |
| DCBM-GDINO (Ours) ↓ | 22.6 | **47.2** | **24.6** |
| DCBM-MASK-RCNN (Ours) ↓ | 22.2 | **44.6** | **22.4** |

using GradCAM.

*Table 3.* **Adaptation to other domains.** Top-1 accuracy reported for MiT-States and ClimateTV (CLIP ViT L/14). DCBMs perform superior to the linear probe when classifying objects in different states. Social media classification performance differs between DCBM concept generation methods.

| Method | MiT-States | ClimateTV |
|---|---|---|
| Linear Probe ↑ | 37.3 | 84.5 |
| Zero-Shot ↑ | 48.1 | 69.7 |
| DCBM-SAM2 (Ours) ↑ | 42.8 | 85.6 |
| DCBM-GDINO (Ours) ↑ | 43.3 | 81.8 |
| DCBM-MASK-RCNN (Ours) ↑ | 43.2 | 87.9 |

DCBMs perform within 6% of the linear probe for all datasets, with superior performance on the fine-grained bird classification dataset CUB (Wah et al., 2011). While CIFAR-10 and CIFAR-100 performance is comparable between CBM approaches, DCBM performance on the large-scale dataset ImageNet and Places365 is slightly subpar compared to other methods. For all concept proposal generation methods (SAM2, GDINO, and MASK-RCNN), DCBM's performance is comparable, indicating that the foundation model can be chosen in accordance with the given application. Appendix F contains results for other CLIP backbones.

### 4.2.2. GENERALIZATION OF DCBM CONCEPTS

We evaluate trained CBM models to understand its generalization capabilities in out-of-distribution domains (ood). To this end, we evaluate CBM models, ours and DN-CBM (Rao et al., 2024), trained on ImageNet on ImageNet-R (Hendrycks et al., 2021) which contains 200 ImageNet classes in various renditions (e.g., embroidery, painting, comic). To ensure a fair comparison, we report the error rates on the same classes within ImageNet (IN-200). Table 2 shows superior ood generalization capabilities of the DCBM model compared to DN-CBM. In contrast to standard IN-R evaluations (Hendrycks et al., 2021), which train the model exclusively on the classes evaluated in IN-R, we re-use the DCBM trained on all 1,000 ImageNet classes. DCBMs trained exclusively on concepts of the 200 IN-R classes are expected to perform even better, given that the generated concepts would better match the target classes and avoid confusion with the 600 classes not contained in ImageNet-R.

Our ood evaluation shows that DCBM exhibits consistently lower error rates on IN-R and a lower gap between IID and OOD in comparison to the task-agnostic DN-CBM (Rao et al., 2024). In Section 4.4, we further validate DCBM's interpretability by analyzing concept activations in test images

We have designed DCBMs to be data efficient and evaluate their performance on novel dataset. In Section 4.1, we introduced MiT-States (Isola et al., 2015) and ClimateTV (Prasse et al., 2023), on which we evaluate the generalization of DCBMs. Table 3 presents the top-1 accuracy of the DCBM prediction, linear probe, and zero-shot classification, all trained on the CLIP ViT L/14 backbone. On both datasets, DCBMs outperform the linear probes. Surprisingly, the zero-shot classification on MiT-States achieves the highest accuracy. The prediction of classes independent of their state appears to be a challenging task. While the different DCBM approaches perform similarly on MiT-States, their performance differs greatly on ClimateTV, with segmentation models' concepts achieving the highest accuracy. Results for awa2 and CelebA can be found in Appendix F.2 along with results for BotCL's subset of ImageNet. We demonstrate that DCBM can capture diverse and complex visual and semantic relationships, showing that it can effectively be applied to other datasets.

### 4.3. DCBM interpretability

We designed DCBMs to provide visual insights into model decision-making. In this section, we qualitatively analyze and interpret the various explanations generated for ImageNet, Places365, and CUB predictions. These evaluations

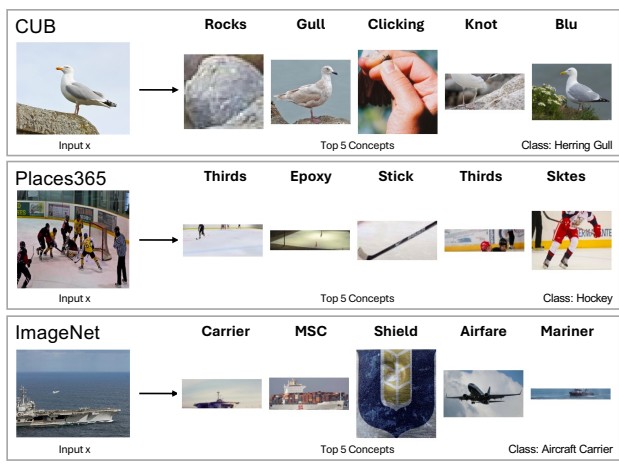

*Figure 3.* **DCBM correct predictions.** Concepts provide insights into the model's embedding space, as it returns visually and/or semantically aligned concepts (SAM2 and CLIP ViT L/14).

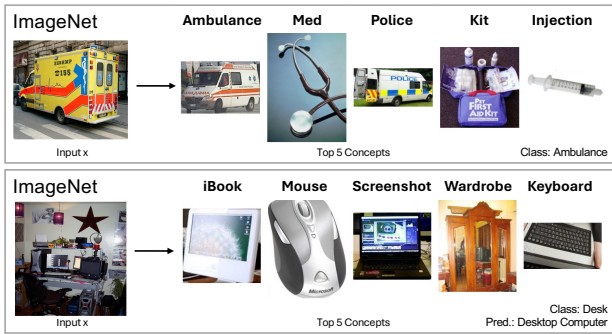

*Figure 4.* **DCBM ambiguous concepts/classes.** Complex visual classes with ambiguous concepts or ambiguous class labels (SAM2 and CLIP ViT L/14).

are conducted on the CLIP ViT L/14 backbone, with SAM2 concept proposals. For each instance, we show the input image along with the five most important concepts. Each concept is displayed alongside its name(as detailed in Section 3.4). We divide this investigation into two parts: in Figure 3, we analyze correct predictions, and in Figure 4, we examine challenging predictions.

The examples demonstrate the correct predictions and showcase concepts from each dataset that closely align with the semantics of the input image (see Section 3.3). The majority of concepts are visually and semantically aligned, as the hockey (Place365) image's most important concepts are *ice*, *(hockey) stick*, *sk(a)tes*, the same holds for the CUB and ImageNet example. Overall, the visual concepts provide sufficient interpretability for the trained CBM's classification, making it applicable to domains that lack textual concepts or rely solely on a vision encoder. Moreover, these explanations highlight the quality of concept naming when incorporated into our framework. For example, in Places365, the concept name thirds corresponds to hockey, where the game is played in three periods of 20 minutes. Given that DCBMs use visual concepts that have been named, the image should always be attributed more weight than the text, as the image-text alignment may be imperfect.

Figure 4 illustrates two examples of challenges arising when using DCBM. The first image is correctly classified using concepts not contained in the image, whereas the second one is incorrectly classified even though the important concepts are contained in the image. For instance, in the case of the ambulance image from ImageNet, concepts such as *med*, *police*, *kit*, or *injection* are not explicitly visible in the input image. This provides insights into the model's embedding space, where a stethoscope is so similar to an ambulance

that it is the second most important concept. In contrast, while the visual concepts for the second instance can be traced back to the input, the predicted class differs from the ground truth, highlighting the inherent ambiguity of the class definition. Further qualitative examples can be found in Appendix G.

## 4.4. DCBM concept investigation

### 4.4.1. CONCEPT CLUSTERING

We evaluate whether concept proposals form cohesive clusters to validate cluster centroid as a visual concept. As shown in Figure 5, different visual representations of the same concept are consistently grouped together. The sample images illustrate strong semantic coherence within clusters across all three datasets, reinforcing the interpretability of the learned concepts. Additionally, the plot illustrates the diversity of concepts captured within each dataset; for instance, ImageNet spans a wide range of object-centered categories, from animals to objects. Depending on the object's complexity, these objects appear as whole instances or parts. In contrast, Places365's scenes are more cluttered and less object-centric. Hence, its concepts include mountain range formations and vegetation. The highly specific CUB dataset contains sub-parts (legs) as well as complex clusters (bird on a tree trunk, vertically). This confirms that foundation models can generate dataset-specific concept proposals suitable for CBM training.

### 4.4.2. CONCEPT LOCALIZATION

To validate DCBM concepts, we measure how well concepts in an explanation can be traced back to the input image using Grad-CAM (Selvaraju et al., 2019) attributions. We employ the Grid Pointing Game (GPG) (Bohle et al., 2021) to assess whether a given class corresponds to the original image when placed in a grid with randomly selected images of different, other classes. Specifically, we include the original

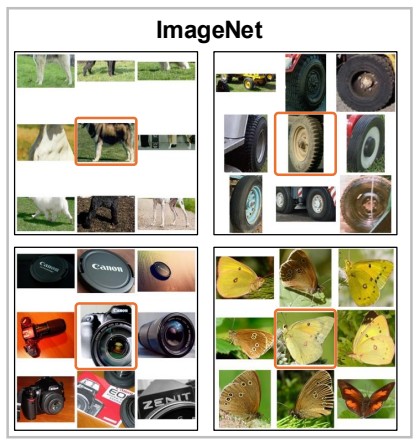
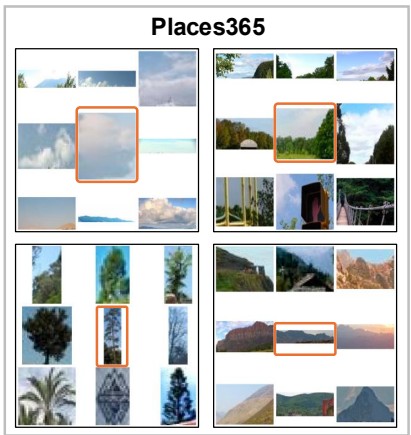
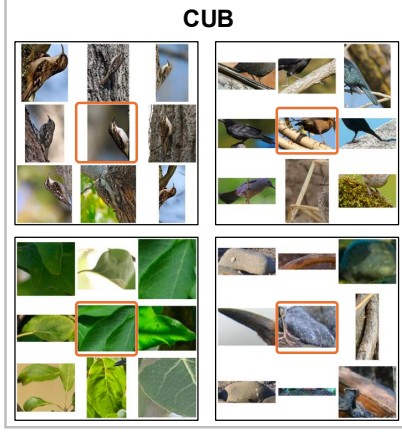

*Figure 5.* **Clustered proposals as concepts.** We show four concept clusters for ImageNet, Places365, and CUB, illustrating that clustering concept proposal embeddings produce semantically coherent clusters across datasets. The center image (orange border) represents the final concept (median image) for each cluster of concept proposals (SAM2 and CLIP ViT L/14).

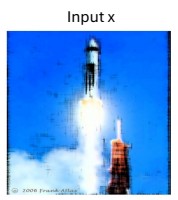
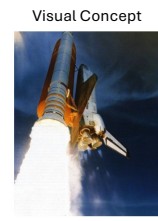
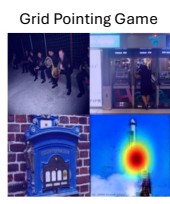

*Figure 6.* **Grid pointing game.** GPG applied to an ImageNet-R instance using a visual concept from ImageNet to demonstrate the effectiveness of our approach in ood (SAM2 and ResNet-50).

*Table 4.* **Grid pointing game - metrics.** Mean values for different metrics are reported for Places365 and CUB (SAM2 and CLIP RN-50).

| Metric | Places365 | CUB |
|---|---|---|
| Gini ↑ | 0.4701 | 0.3608 |
| Percentage ↑ | 0.6519 | 0.3402 |
| Abs ↑ | 0.6551 | 0.3445 |

test image in a 2×2 grid with three randomly chosen images from the same dataset (see Figure 6). We then select the top five visual concepts as classes for Grad-CAM and apply them to each visual concept within the composite 2×2 grid image, analyzing whether activation maps align with the original image's quadrant. This evaluation is conducted on CUB and Places365 test sets using the ResNet-50 backbone. In Table 4, we compute the mean over the entire test set for each of the three metrics: (1) Gini Index, which measures the attribution score dispersion across the grid, (2) relative percentage, which measures how much normalized attribution overlaps with the correct quadrant, and (3) maximum score, which measures whether the highest attribution score is in the correct quadrant. For all metrics, higher values indicate a better localization within the specific test image. The GINI index is defined within [0,1], where 0 stands for equal attributions in the entire group and 1 indicates perfect localizability. Percentage and absolute scores are lower bounded at 0.25, where the attributions would be randomly divided between the 4 images in the grid.

The results validate that DCBM concepts can be localized within test images. While Places365 shows higher attribu-

tion alignment (e.g., Gini: 0.4701 vs. 0.3608, Abs: 0.7028 vs. 0.4988), CUB also demonstrates better than random coherence. Overall, the lower numbers in CUB, a dataset for fine-grained classification, are also to be expected since several concepts are usually shared between classes (e.g., different bird species have similar beaks) and are thus activated along with the other class. The consistent performance across metrics validates our method's ability to map visual concepts to relevant image regions.

## 5. Discussion

With DCBMs, we create a simple and data-efficient framework for model interpretability, which can easily be adapted to new datasets. Using segmentation and detection foundation models, we retrieve image-inherent concepts in the visual domain. For any prediction, we can trace their concepts' activations back to the test image. DCBM's accuracy is in all experiments within 5% of the linear probe, indicating that its interpretable predictions are comparable between common and uncommon CBM datasets. Based on these findings, we expect to see similar behavior for other datasets. DCBM's applicability is limited by two factors, the suitability of a segmentation foundation model and the expressiveness of

the CLIP embedding space for a given use case. Further advances in vision-language models, segmentation, and detection models can enhance results, as new methods can easily be integrated into the DCBM framework.

In comparison to other CBMs, we achieve the highest performance on CUB (Wah et al., 2011). The task-specific nature of our concepts appears well-suited for fine-grained classification. The task-agnostic DN-CBM (Rao et al., 2024) does not report their CUB performance, but their outlook proposes pre-training on larger datasets to enhance fine-grained classification. As of now, we steer the concept generation process solely by clustering similar concept proposals into a single cluster. Our ablations of removing small or large concepts show a small influence on CBM performance. Moreover, concept proposal steering using the promptable GroundingDino results in a lower number of concept proposals. Given that accuracy is comparable to steering-free methods such as SAM2, we can assume that the steered concept set contains equally relevant concepts. Additionally, DCBMs allow the removal of undesired concepts using a text interface, making it adjustable to spurious correlations within a dataset's context.

Qualitative concept analysis (Section 4.3) shows that DCBMs' concepts contain image- and part-level concepts of the dataset's classes. DCBM's interpretations highlight spurious correlations within the CLIP embedding space by basing decisions on semantically close concepts that are not present in the test image. While spurious correlations can be observed in all CLIP CBMs (Menon & Vondrick, 2023; Oikarinen et al., 2023; Panousis et al., 2023; Rao et al., 2024; Yang et al., 2023), DCBMs allow for the investigation of visual spurious correlation within dataset-specific concepts. Attribution methods such as GradCAM can visualize these semantic similarities between visually different concepts. We utilize DCBM as an inspection tool and can exclude undesired concepts from CBM training. Ensuring concept grounding using slot attention as in(Wang et al., 2023) or an image tagging model such as RAM (Zhang et al., 2024b) can be an automated measure against spurious correlations.

The current concept set contains the clustered crops of segments or object detections, a simple and efficient approach. Regularization the concept selection can incorporate concept diversity concerning hierarchy. Previous works have defined a threshold for image coverage (Zhu et al., 2024) to determine the number of concepts. We believe that detailed investigations can reduce the number of clusters without restricting their diversity and granularity. Additionally, other embedding spaces can be evaluated to discuss inter-model differences. CBM sparsity can be further increased to increase model interpretability (Schrodi et al., 2024).

A possible next step can be the extension of DCBMs to have post-hoc interpretability by learning a mapping from model features to concepts. Besides image-concept similarity, we argue for integrating the GPG results into the mapping to ensure having the most suitable concept for each object in the image. In all cases, we strongly argue for employing a single model at a time to maintain model-specific interpretability. Moreover, we believe it would be interesting to extend DCBMs or CBMs in general to tasks beyond classification. Interpretable regression could be learned by integrating GPG into the prediction pipeline, assuming that the concept represents the largest extent of a given concept and that concept activations would be less when the extent is less. One use case for interpretable regression would be severity prediction for skin lesions.

Lastly, efficiency is a strength of DCBMs since they do not require general (Rao et al., 2024) or dataset-specific pre-training (Schrodi et al., 2024). Moreover, the concept proposal set is created using only 50 images per class; fewer images may be used as a trade-off for accuracy.

## 6. Conclusion

In this work, we present DCBM, a novel approach to CBMs that uses data-efficient concepts entirely in the visual domain. DCBMs employ a simple and intuitive approach to deriving concepts by using segmentation or detection foundation models. Clustering concept proposals, reduces the different representations of concepts to a single prototypical one, which is used for CBM training and evaluation. DCBM's efficient and unrestricted adaptability to other datasets highlights its potential as a CBM framework for any dataset. Our approach generalizes well across domains and datasets, demonstrating versatility with respect to concept proposal methods like SAM2, GroundingDINO, and MASK-RCNN. Given that no pre-training is required, DCBMs can provide interpretations for a new dataset in under one hour (depending on dataset size).

## Impact Statement

This paper presents research that aims to ease the access to classification interpretability. Moreover, we want to highlight that misleading classifications may reduce trust in AI applications rather than increase it. These impacts can be observed within any CBM and are not specific to DCBM.

## Acknowledgements

This work is partially supported by the BMBF (Federal Ministry of Education and Research) project 16DKWN027b Climate Visions and by the German Federal Ministry for Economic Affairs and Climate Action of Germany (BMWK). All experiments were run on University of Mannheim's server.

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

# A. Framework

We describe the DCBM framework in the main paper. For better understanding, we provide the framework with pseudocode in Algorithm 1 and introduce notation to this end.

An object $O$ can be represented as a combination of concepts from a global concept pool $\mathcal{C}$. Let $C_i \in \mathcal{C}$ denote individual concepts, and let each object be characterized by a subset of these concepts with corresponding weights.

$$O = \sum_{i=1}^{|\mathcal{C}|} w_i C_i, \quad w_i \geq 0$$

where: $C_i \in \mathcal{C}$ represents a concept from the global pool, $w_i$ denotes the contribution of concept $C_i$ to the image, $C_i$ is located in the image $I$, $R_i \subseteq I$ is the region of the image where concept $C_i$ is present,

Further, we assume visual concepts to be spatially localized in images and therefore conjecture that data efficient concept extraction should build upon image segments or regions rather than entire images. Therefore, we generate the global concept pool $\mathcal{C}$ by segmentation or detection foundation models, which are then cropped out and used as concept proposals $s_i$. All $s_i \in \mathcal{S}$ are then clustered into the global concept set $\mathcal{C}$.

---

**Algorithm 1** DCBM Framework

---

1: **Input:** $\mathcal{D}$ (Training dataset), $\mathcal{T}$ (Text corpus), $n$ (Number of images per class), $\Omega$ (Segmentation model), $f$ (Vision encoder), $f_{\text{text}}$ (Text encoder), $k$ (Number of clusters), $a_{\min}$ and $a_{\max}$ (Minimum and maximum area thresholds for segmentation).

2: **1. Concept Proposal Generation**

3: $D_n \leftarrow \{\text{randomly select } n \text{ images from each class in } \mathcal{D}\}$

4: $\mathcal{S} \leftarrow \{\Omega(x) \mid x \in D_n\}$ {Apply segmentation to each image}

5: $\mathcal{S} \leftarrow \{\text{crop}(x, s_i^l) \mid x \in D_n, \ s_i^l \in \mathcal{S}\}$ {Extract bounding boxes based on segmentation}

6: $\mathcal{S} \leftarrow \{s_i^l \in \mathcal{S} \mid \text{area}(s_i^l) \in [a_{\min}, a_{\max}]\}$ {Filter segments by area}

7: **2. Concept Creation**

8: $\mathcal{S}_{\text{enc}}^d \leftarrow f(\mathcal{S})$ {Encode segmented concepts with vision encoder}

9: $\mathcal{D}_{\text{enc}}^d \leftarrow f(\mathcal{D})$ {Encode entire dataset}

10: $\mathcal{T}_{\text{enc}}^d \leftarrow f_{\text{text}}(\mathcal{T})$ {Encode text corpus}

11: ids $\leftarrow \text{cluster}(\mathcal{S}_{\text{enc}}^d, k)$ {Cluster encoded segments into $k$ clusters}

12: $\mathcal{C} \leftarrow \{\text{centroid}(\mathcal{S}_{\text{enc}}^d, \text{ids} = i) \mid i = 1, \ldots, k\}$ {Compute centroids for each cluster}

13: **3. Concept Bottleneck Model**

14: $A \leftarrow \frac{\langle \mathcal{D}_{\text{enc}}^d, c_j \rangle}{\|c_j\|_2^2}, \quad$ for $j = 1, \ldots, k$ {$A$ represents the activation of concepts in $\mathcal{D}$}

15: $\text{DCBM} \leftarrow \text{fit}(A, y \mid y \in \mathcal{D})$ {Train linear model with ground truth}

16: **4. Concept Naming (optional)**

17: $sim \leftarrow \text{cos\_sim}(\mathcal{T}_{\text{enc}}^d, \mathcal{C})$ {Compute similarity between textual and visual concepts}

18: $t \leftarrow \text{zeros}(|\mathcal{T}_{\text{enc}}^d|)$ {Initialize $t$ as a zero vector with the same length as $\mathcal{C}$}

19: **for** $c_j \in \mathcal{C}$ **do**

20: $\quad t_j \leftarrow \arg\max(sim_j)$ {Get the index of the maximum similarity in the $j$-th row}

21: **end for**

22: $\mathcal{C}_{\text{named}} \leftarrow \{\mathcal{T}[t_j] \mid t_j \in \{1, 2, \ldots, |\mathcal{T}|\}\}$ {Named concepts with descriptions}

23: **Output:**

24: DCBM, $\mathcal{C}, \mathcal{C}_{\text{named}}$

---

The underlying feature extraction method is denoted by $\Omega$ with the hyperparameter settings as $hp$. Possible choices are the foundation models Segment Anything 1 & 2 (Kirillov et al., 2023; Ravi et al., 2025), GroundingDino (Liu et al., 2024), MaskRCNN (He et al., 2017), and DETR (Carion et al., 2020). We compare the segmentation models, which require no input, to the open-set detection method GroundingDino, in which we use various concept sets from the literature as input. We construct a concept proposal subset, $\mathcal{D}$, by selecting a configurable number $n$ of random training images per class. This has the goal of mitigating the risk of overfitting while increasing efficiency since not all training images are used for training. We set $n = 50$ for the main paper, for CUB, which contains fewer samples, we include all available images in $\mathcal{D}$. In Appendix A.1 we discuss DCBM's data-efficiency in detail.

Each image $i$ in the subset $\mathcal{D}$ is segmented by $\Omega$ into $l$ concept proposals, denoted as $s_i^l$. The number of concept proposals varies between images and is determined by the hyperparameter settings $hp$. We crop the concept proposal's bounding box and add the resulting sub-image to concept proposal set as $\mathcal{S}$. The removal of background information is ablated in Appendix D.1. To refine the concept proposals, image parts that are below or above a given size threshold can be excluded from $\mathcal{S}$, as evaluated in Appendix E.2.

### A.1. Data-efficiency

DCBM is highly data-efficient in the concept generation phase and equally efficient as other models in the CBM training phase.

In Table 5 we show the differences to another data-driven CBM, i.e., DN-CBM (Rao et al., 2024). DCBM does not require general pre-training, but it can be directly employed for the dataset in question. This makes DCBM an effective tool in single-dataset settings.

|  | DN-CBM | DCBM - ImageNet |
| --- | --- | --- |
| Dataset size | 3,300k image-caption pairs (CC3M) | 50k images (50 imgs/class) |
| Add. memory capacity | 850 GB (assuming 256x256px) | 6 GB |
| No extra data required | $\times$ | $\checkmark$ |

*Table 5.* CBM training preparation: DN-CBM (Rao et al., 2024) vs DCBM (ours).

We have set $n = 50$, i.e., selected only 50 images per class for the concept proposal generation. This subset is sufficiently large and diverse, as Table 6 shows. Using an alternative subset or the combination of the existing and new subset does not have an impact on the CBM accuracy. CUB is not included in the table as it has less than 50 images per class and thus the entire dataset is used for concept proposal generation.

|  | *# imgs / class* | ImageNet | Places365 | Cifar10 | Cifar100 |
| --- | --- | --- | --- | --- | --- |
| s1 (main paper) | *50* | 77.4 | 52.2 | 97.5 | 85.3 |
| s2 (new) | *50* | 77.5 | 52.2 | 97.6 | 85.4 |
| s1+s2 | *100* | 77.1 | 52.1 | 97.7 | 85.5 |

*Table 6.* Performance evaluation of subset selection

Data-efficiency can be further increased by using fewer training samples at the cost of slight performance degradation, as shown in Appendix E.1. Moreover, reducing the number of training data points for CBMs is feasible and results in only a modest performance degradation, as shown in Table 7. This relatively minor trade-off highlights the potential applicability of DCBMs in data-scarce environments.

| *# imgs / class* | ImageNet | Places365 | Cifar10 | Cifar100 |
| --- | --- | --- | --- | --- |
| all | 77.4 | 52.2 | 97.5 | 85.3 |
| 50 imgs / class (SAM2) | 75.5 | 46.1 | 91.2 | 79.1 |
| 50 imgs / class (GDINO) | 75.0 | 46.2 | 93.1 | 79.4 |
| 50 imgs / class (MASK-RCNN) | 75.5 | 46.5 | 94.8 | 80.3 |

*Table 7.* Performance evaluation when reducing the number of training samples to 50 images per class (ViT-L/14).

# B. Dataset overview and details

## B.1. Ablations

We ablate on three datasets: ImageNette (Howard, 2019a), ImageWoof (Howard, 2019b), and CUB (Wah et al., 2011). The first two datasets are both subsets of ImageNet (Deng et al., 2009). ImageNette contains easier-to-classify categories like tench, English springer, and church. ImageWoof images focus on dog breeds, curated for fine-grained classification tasks. The Caltech-UCSD Birds 200 dataset is designed to identify fine-grained bird species with detailed annotations. For the datasets which do not have a test split, we use the validation split for testing and create a new split, i.e., , 10% of train set, for validation. Consequently, the train split comprises only 90% of the original train images (see Table 8).

*Table 8.* **Ablation datasets.** Overview of the datasets used for ablation (ImageWoof, ImageNette, and CUB-200-2011).

| Dataset | Classes | Images %(train / val / test) |
|---|---|---|
| ImageNette | 10 | 13,000 (70 / 30 / 0) |
| ImageWoof | 10 | 12,000 (70 / 30 / 0) |
| CUB-200-2011 | 200 | 11,788 (50 / 50 / 0) |

## B.2. Benchmark analysis

For the primary analysis, we compare five datasets: Imagenet (Deng et al., 2009), Places365 (Zhou et al., 2017), CUB (Wah et al., 2011), cifar10 and cifar100 (Krizhevsky et al., 2009). For ImageNet, we report on the validation split. Thus, we use it as our test set. Again, we create a small validation set to select the hyperparameters on an independent dataset in situations where we cannot access three labeled splits (see Table 9).

*Table 9.* **Main datasets.** Overview of the datasets used for benchmark experiments.

| Dataset | Classes | Images (train / val / test) |
|---|---|---|
| ImageNet | 1,000 | 1,431,167 (90 / 3 / 7) |
| Places365 | 365 | 1,839,960 (98 / 2 / 0) |
| CUB-200-2011 | 200 | 11,788 (50 / 50 / 0) |
| CIFAR-10 | 10 | 60,000 (83 / 0 / 17) |
| CIFAR-100 | 100 | 60,000 (83 / 0 / 17) |

## B.3. Additional datasets

We use five datasets which are novel or rare to the CBM research and summarize them in Table 10: MiT-States (Isola et al., 2015), Climate-TV (Prasse et al., 2023), ImageNet-R (Hendrycks et al., 2021), awa2 (Xian et al., 2018), and CelebA (Liu et al., 2015). MiT-States contains images of 245 object classes in combination with 115 attributes. This allows the assessment of concept recognition in previously unseen shapes or forms, as the test set contains 50% seen and 50% unseen versions of objects, e.g., ripe tomato, unripe tomato, moldy tomato. We evaluate the accuracy of the object class. Thus, we use this dataset to assess whether we can detect an object independent of its state. We use the train-val-test split (30k-10k-13k) introduced by (Purushwalkam et al., 2019). ClimateTV contains social media images of climate change and includes a diverse range of images. We have created a balanced set for the animals-superclass, which contains 11 animal classes, including "no animals". This allows the assessment of real-life images, which are, by design, messier than curated datasets. We employ animals with attributes in the drop-in version 2 (Xian et al., 2018) consisting of animal images labeled with 85 numeric attributes. We have run DCBM on AwA using the standard 50:50 train and test split. We created the val set by randomly selecting 10% of train samples. Lastly, we employ CelebA (Liu et al., 2015), which contains face photographs of 10,177 celebrities, in which 5 landmark locations and 40 binary attribute annotations are provided for each image. This dataset is designed for face landmark detection and is transformed as described by Zhang et al. (2024a), using a 70 : 10 : 20

(train:val:test) split. We randomly sample every 12th image as done by past research (Zhang et al., 2024a). In order to move towards a classification datasets, the 8 most balanced attributes are taken and all possible combinations are used as class labels, thus resulting into 256 possible classes. The most balanced attributes are: Attractive, Mouth_Slightly_Open, Smiling, Wearing_Lipstick, High_Cheekbones, Male, Heavy_Makeup, and Wavy_Hair. Given that some combinations of attributes are highly unlikely, we exclude classes with less than 10 samples, thus resulting in 110 actual classes.

*Table 10.* **Additional datasets.** Overview of the additional dataset (MiT-States, ClimateTV, and ImageNet-R).

| Dataset | Classes | Images %(train / val / test) |
|---|---|---|
| MiT-States | 245 | 53,743 (57 / 19 / 24) |
| ClimateTV | 11 | 660 (80 / 0 / 20) |
| AWA2 | 50 | 37,322 (16,789/ 1,863/ 18,670) |
| CelebA | 110 | 16,651 (11,603 / 1,602 / 3,446) |
| ImageNet-R | 200 | 30,000 (0 / 100 / 0) |

ImageNet-R consists of 200 ImageNet classes and is generally used to evaluate ood performance of a model trained on ImageNet. The "R" stands for renditions of which the dataset contains cartoons, graffiti, embroidery, graphics, origami, paintings, and many more (Hendrycks et al., 2021). This task is simplified by training the model exclusively on the 200 classes of ImageNet-R. In our case, however, we report the model's performance when trained on all ImageNet classes.

## C. Concept proposal generation

We provide the code for all segmentation methods employed and make it publicly available. Table 11 provides an overview of the segmentation models' hyperparameters, which are unchanged for MaskRCNN and DETR compared to the pre-trained models. Hyperparameters were set to retrieve concept proposals that break the image into sub-parts. For GroundingDino, we evaluated two thresholds to compare whether more, noisier or fewer, cleaner concept proposals performed better.

*Table 11.* **Hyperparameters of segmentation models.** We only report model hyperparameters if changed compared to standard setting.

| Segmentation model | Hyperparameters |
|---|---|
| SAM | points_per_side = 64, pred_iou_thresh = 0.88, stability_score_thresh = 0.95, box_nms_thresh = 0.5, min_mask_region_area = 500 |
| SAM2 | points_per_side = 64, pred_iou_thresh = 0.88, stability_score_thresh = 0.95, box_nms_thresh = 0.5, min_mask_region_area = 500 |
| GDINO | box_threshold = [0.35, 0.25], text_threshold = 0.25 |

### C.1. Prompts

For the promptable GroundingDINO model, we assess the efficacy of several prompts w.r.t. CBM performance. To this end, we use attribute or part labels, which have been published in other contexts, thus avoiding manual or LLM-based concept set generation. We use the part annotations for CUB (Wah et al., 2011), the attributes for Animals with Attributes (GDINO Awa) (Lampert et al., 2009), and standard part-labels from Part-ImageNet (He et al., 2022) and GDINO Pascal-PARTS (Chen et al., 2014).

# D. Ablations

We conduct ablation studies to evaluate DCBM's performance under various hyperparameters and configurations. These studies are performed on the ImageNette, ImageWoof, and CUB to identify settings that yield strong performance across all three datasets. The identified configurations are used for the main experiments presented in Section 4. Each DCBM was trained with 1024 clusters, using K-Means clustering with the median of each cluster as a concept, a learning rate of $1e-4$, and sparsity regularization set to $1e-4$. We explore a broader range of segmentation and object detection models beyond those discussed in the main paper, such as SAM and extended prompts for GDINO. The "Low" label indicates a low threshold for the corresponding object detection configuration. What differentiates these experiments from those in the main paper is the use of PCA, reducing the dimensionality of concept proposals to 100. This approach accelerates clustering and enables more efficient experimentation. Additionally, we investigate the impact of PCA on the performance of DCBM.

## D.1. Background removal

The background influences embedding the image regions used as concepts for the segmentation methods employed. To assess this effect, we examine accuracies in Table 12 using three different CLIP embeddings: ResNet-50, ViT B/16, and ViT L/14 — while excluding the background concept, based on ImageNette, ImageWoof, and CUB. We report SAM, SAM2, Mask R-CNN, and DETR results. Each model was trained with a learning rate of $1e-4$, a sparsity parameter $\lambda$ of $1e-4$, 128 clusters using median and k-means, and 50 images per class within the DCBM framework.

*Table 12.* **Background performance influence.** Performance comparison with (w/) and without (w/o) background on the ablation datasets.

| Model | CLIP ResNet-50 | | | CLIP ViT-B/16 | | | CLIP ViT-L/14 | | |
|---|---|---|---|---|---|---|---|---|---|
| | ImageNette | ImageWoof | CUB | ImageNette | ImageWoof | CUB | ImageNette | ImageWoof | CUB |
| SAM w/o | 98.55 | 90.96 | 57.04 | 99.57 | 93.66 | 71.68 | 99.87 | 95.42 | 80.03 |
| SAM w/ | 98.50 | 90.81 | 57.62 | 99.54 | 93.79 | 73.44 | 99.80 | 97.78 | 79.58 |
| SAM2 w/o | 98.73 | 91.70 | 59.89 | 99.52 | 94.38 | 73.61 | 99.87 | 95.75 | 80.98 |
| SAM2 w/ | 98.78 | 91.68 | 61.44 | 99.57 | 94.30 | 75.49 | 99.85 | 95.80 | 82.21 |
| MaskRCNN w/o | 98.65 | 91.93 | 64.62 | 99.59 | 94.38 | 77.60 | 99.80 | 95.70 | 83.60 |
| MaskRCNN w/ | 98.68 | 91.89 | 65.46 | 99.59 | 94.60 | 77.61 | 99.82 | 95.65 | 83.32 |
| DETR w/o | 98.65 | 91.80 | 64.17 | 99.46 | 94.45 | 76.84 | 99.82 | 95.75 | 82.57 |
| DETR w/ | 98.73 | 92.03 | 61.18 | 99.54 | 94.58 | 76.91 | 99.80 | 95.78 | 82.52 |

## D.2. PCA

To assess the impact of different hyperparameter settings within the DCBM framework, we analyze clustering similarities across various values of $k$ in K-means and compare these results across different CLIP backbone models in Figure 7. Our primary focus is on the CUB dataset, as its exclusive focus on bird images introduces ambiguity in clustering compared to other datasets. Additionally, in Figure 8, we examine the effect of applying PCA with 100 components to the segment embeddings before clustering to determine whether this transformation yields similar clustering outcomes. For this evaluation, we use the NMI (Normalized Mutual Information) metric, where an NMI score close to 1 indicates high similarity, meaning that the two clustering approaches capture comparable groupings or structures. In contrast, a score near 0 suggests minimal alignment, indicating substantially different clustering results.

*Table 13.* **Impact of PCA on CUB dataset.** Accuracy comparison of DCBM with and without PCA (100 components) evaluated on the CUB dataset.

|  | ResNet-50 | CLIP ViT-B/16 | CLIP ViT-L/14 |
|---|---|---|---|
| DCBM-SAM2 (Ours) w | 61.2 | 75.1 | 81.9 |
| DCBM-SAM2 (Ours) w/o | 61.4 | 75.3 | 81.8 |
| DCBM-GDINO (Ours) w | 58.9 | 73.9 | 81.1 |
| DCBM-GDINO (Ours) w/o | 59.0 | 74.1 | 81.3 |
| DCBM-MASK-RCNN (Ours) w | 64.6 | 76.8 | 82. |
| DCBM-MASK-RCNN (Ours) w/o | 64.6 | 76.7 | 82.4 |

As shown in Figure 7, the B16, L14 combination consistently achieves the highest NMI scores. Additionally, as the number of clusters $k$ increases, the NMI scores improve in all segmentation techniques and backbone combinations, indicating that larger cluster sizes capture more meaningful distinctions in the data. These results suggest concept clusters remain similar across different CLIP backbones, with minimal unique variation among specific backbone combinations.

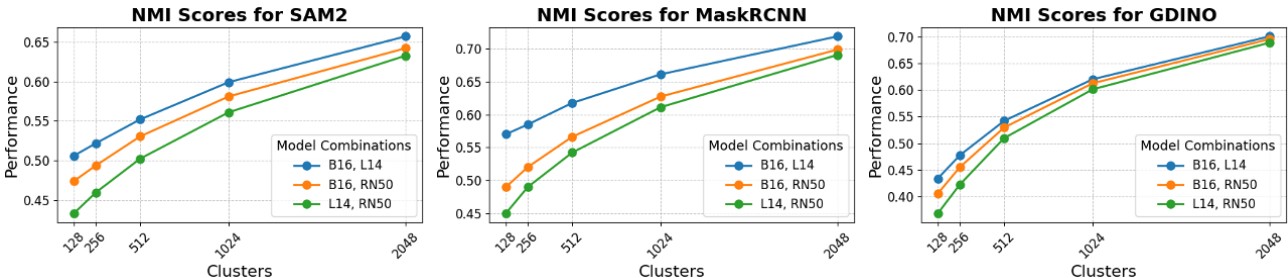

*Figure 7.* **Backbone NMI scores for the CUB dataset.** NMI scores for three CLIP model combinations: B16, L14 (blue), B16, RN50 (orange), and L14, RN50 (green), showing clustering performance across different backbones across cluster sizes (128, 256, 512, 1024, 2048).

Similarly, Figure 8 illustrates the impact of PCA on cluster similarity. While PCA reduces dimensionality and speeds up the clustering process, the clusters formed remain broadly consistent with those generated without PCA. This suggests that PCA minimally affects the conceptual alignment within CBMs, preserving the overall clustering structure. Furthermore, as demonstrated in Table 13 for CUB, this limited influence extends to the overall performance of DCBMs.

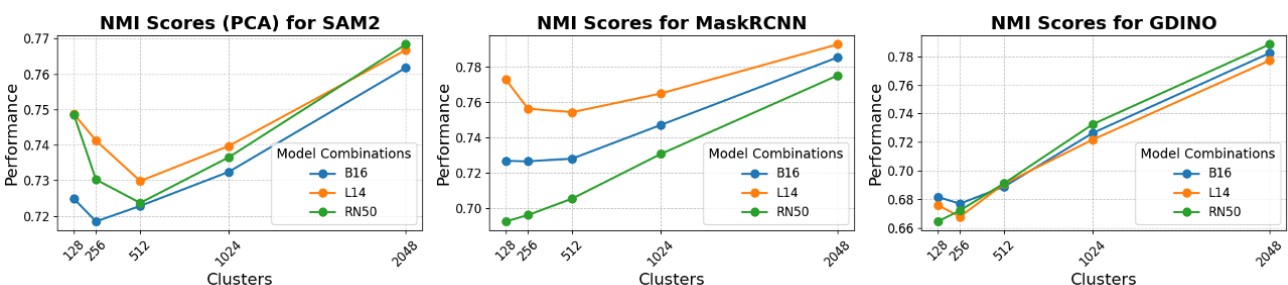

*Figure 8.* **PCA NMI scores for the CUB dataset.** NMI scores illustrate clustering consistency with and without PCA preprocessing (100 components) for each CLIP backbone across varying cluster sizes (128, 256, 512, 1024, 2048).

## D.3. Clustering algorithm

We evaluate two clustering algorithms in the main ablation study: hierarchical clustering and K-means. The results, presented in Table 14, are based on a consistent configuration where the median centroid is used, and the number of clusters $k$ is fixed at 1024 across all datasets. While the performance scores of both methods are generally comparable, K-means slightly outperforms hierarchical clustering in most cases. Consequently, we use K-means for the main experiments.

*Table 14.* **Clustering method comparison.** Performance comparison of different clustering algorithms (Hierarchical and K-Means), both with $k$ set to 1024.

| Model | Clustering Method | CLIP ResNet-50 | | | CLIP ViT-B/16 | | | CLIP ViT-L/14 | | |
|---|---|---|---|---|---|---|---|---|---|---|
| | | ImageNette | ImageWoof | CUB | ImageNette | ImageWoof | CUB | ImageNette | ImageWoof | CUB |
| SAM | Hierarchical | 98.60 | 90.74 | 61.15 | 99.52 | 93.38 | 73.27 | 99.82 | 95.65 | 82.11 |
| | K-means | 98.52 | 90.86 | 57.62 | 99.57 | 93.74 | 73.44 | 99.82 | 95.90 | 80.03 |
| SAM2 | Hierarchical | 98.75 | 92.11 | 61.15 | 99.62 | 94.45 | 75.39 | 99.85 | 95.80 | 82.12 |
| | K-means | 98.72 | 92.19 | 61.44 | 99.59 | 91.60 | 75.60 | 99.85 | 95.70 | 81.88 |
| DETR | Hierarchical | 98.73 | 92.03 | 63.96 | 99.54 | 94.53 | 76.82 | 98.77 | 95.85 | 82.31 |
| | K-means | 98.65 | 92.21 | 64.27 | 99.54 | 94.60 | 77.01 | 99.80 | 95.80 | 82.64 |
| MaskRCNN | Hierarchical | 98.68 | 92.03 | 65.41 | 99.57 | 94.35 | 77.53 | 99.85 | 95.78 | 83.21 |
| | K-means | 98.73 | 91.86 | 65.46 | 99.60 | 94.60 | 77.70 | 99.85 | 95.65 | 83.33 |
| GDINO Awa | Hierarchical | 98.57 | 91.27 | 58.54 | 99.41 | 93.99 | 74.13 | 99.87 | 95.70 | 81.36 |
| | K-means | 98.50 | 91.17 | 57.97 | 99.46 | 94.04 | 73.63 | 99.89 | 95.60 | 81.12 |
| GDINO Awa Low | Hierarchical | 98.60 | 90.81 | 57.84 | 99.44 | 94.15 | 73.92 | 99.87 | 95.62 | 81.17 |
| | K-means | 98.57 | 90.91 | 65.43 | 99.41 | 94.17 | 77.70 | 99.87 | 95.67 | 83.09 |
| GDINO Partimagenet | Hierarchical | 98.47 | 91.30 | 59.30 | 99.39 | 94.12 | 74.30 | 99.80 | 95.54 | 81.14 |
| | K-means | 98.47 | 91.27 | 59.63 | 99.44 | 94.17 | 74.70 | 99.82 | 95.62 | 81.52 |
| GDINO Partimagenet Low | Hierarchical | 98.55 | 90.76 | 58.99 | 99.46 | 94.25 | 74.44 | 99.82 | 95.55 | 81.14 |
| | K-means | 98.52 | 90.99 | 59.39 | 99.52 | 94.17 | 74.20 | 99.85 | 95.83 | 81.22 |
| GDINO Pascal | Hierarchical | 98.60 | 90.81 | 58.77 | 99.44 | 94.15 | 73.66 | 99.87 | 95.62 | 80.74 |
| | K-means | 98.57 | 90.91 | 58.80 | 99.41 | 93.17 | 73.73 | 99.87 | 95.67 | 81.00 |
| GDINO Pascal Low | Hierarchical | 98.62 | 90.84 | 58.37 | 99.49 | 94.10 | 73.97 | 99.85 | 95.75 | 81.03 |
| | K-means | 98.62 | 90.89 | 58.25 | 99.52 | 93.89 | 73.78 | 99.85 | 95.75 | 81.15 |
| GDINO Sun | Hierarchical | 98.09 | 83.10 | 58.85 | 99.16 | 88.90 | 73.70 | 99.62 | 93.40 | 80.95 |
| | K-means | 98.16 | 82.97 | 58.85 | 99.18 | 88.78 | 73.77 | 99.62 | 93.54 | 81.00 |
| GDINO Sun Low | Hierarchical | 98.60 | 90.94 | 58.94 | 99.47 | 93.79 | 74.42 | 99.82 | 95.88 | 81.36 |
| | K-means | 98.60 | 90.94 | 59.00 | 99.47 | 94.46 | 74.54 | 99.80 | 95.88 | 81.51 |

Additionally, we tested DBSCAN and HDBSCAN. However, these methods required significant hyperparameter tuning to achieve meaningful clusters. In contrast, K-means and hierarchical clustering required only the definition of $k$. We leave this analysis for future work.

## D.4. Centroid selection

After clustering the concept proposals $\mathcal{S}$ into $k$ clusters, a centroid is generated for each cluster to represent it. We evaluate the impact of using two different methods to compute centroids: the mean of the embeddings within a cluster and the median embedding, as shown in Table 15. This analysis is applied to both integrated clustering techniques while keeping all other configurations consistent with the standard settings. Overall, the K-means configuration using the median centroid performs the best across the three datasets. While there are specific scenarios where other combinations yield better results, we adopt this standard setting for simplicity following this investigation.

*Table 15.* **Centroid selection: mean vs. median.** This figure illustrates the comparative analysis of centroid selection methods — mean and median — on hierarchical and K-means clustering performance.

| Model | Clustering Method | Centroid | CLIP ResNet-50 | | | CLIP ViT-B/16 | | | CLIP ViT-L/14 | | |
|---|---|---|---|---|---|---|---|---|---|---|---|
| | | | ImageNette | ImageWoof | CUB | ImageNette | ImageWoof | CUB | ImageNette | ImageWoof | CUB |
| SAM | Hierarchical | Mean | 98.45 | 89.62 | 56.78 | 99.44 | 92.95 | 68.66 | 99.75 | 95.39 | 80.34 |
| | Hierarchical | Median | 98.60 | 90.74 | 61.15 | 99.52 | 93.38 | 73.27 | 99.82 | 95.65 | 82.11 |
| | K-means | Mean | 98.42 | 89.67 | 48.64 | 99.46 | 93.03 | 69.33 | 99.77 | 95.37 | 76.49 |
| | K-means | Median | 98.52 | 90.86 | 57.62 | 99.57 | 93.74 | 73.44 | 99.82 | 95.90 | 80.03 |
| SAM2 | Hierarchical | Mean | 98.73 | 92.03 | 56.78 | 99.57 | 94.42 | 73.11 | 99.85 | 95.75 | 80.34 |
| | Hierarchical | Median | 98.75 | 92.11 | 61.15 | 99.62 | 94.45 | 75.39 | 99.85 | 95.80 | 82.12 |
| | K-means | Mean | 98.73 | 92.20 | 56.63 | 99.59 | 91.60 | 73.30 | 99.82 | 95.83 | 80.19 |
| | K-means | Median | 98.72 | 92.19 | 61.44 | 99.59 | 91.60 | 75.60 | 99.85 | 95.70 | 81.88 |
| DETR | Hierarchical | Mean | 98.68 | 92.11 | 63.46 | 99.54 | 94.50 | 76.46 | 99.80 | 95.75 | 82.00 |
| | Hierarchical | Median | 98.73 | 92.03 | 63.96 | 99.54 | 94.53 | 76.82 | 98.77 | 95.85 | 82.31 |
| | K-means | Mean | 98.68 | 92.19 | 63.27 | 99.54 | 94.52 | 76.23 | 99.80 | 95.80 | 81.69 |
| | K-means | Median | 98.65 | 92.21 | 64.27 | 99.54 | 94.60 | 77.01 | 99.80 | 95.80 | 82.64 |
| MaskRCNN | Hierarchical | Mean | 98.70 | 91.65 | 63.07 | 99.59 | 94.48 | 76.75 | 99.79 | 95.62 | 82.41 |
| | Hierarchical | Median | 98.68 | 92.03 | 65.41 | 99.57 | 94.35 | 77.53 | 99.85 | 95.78 | 83.21 |
| | K-means | Mean | 98.75 | 91.73 | 63.00 | 99.59 | 94.63 | 77.13 | 99.82 | 95.62 | 82.22 |
| | K-means | Median | 98.73 | 91.86 | 65.46 | 99.60 | 94.60 | 77.70 | 99.85 | 95.65 | 83.33 |
| GDINO Awa | Hierarchical | Mean | 98.57 | 91.14 | 52.54 | 99.44 | 94.02 | 70.59 | 99.85 | 95.78 | 78.60 |
| | Hierarchical | Median | 98.57 | 91.27 | 58.54 | 99.41 | 93.99 | 74.13 | 99.87 | 95.70 | 81.36 |
| | K-means | Mean | 98.62 | 91.07 | 53.00 | 99.41 | 93.97 | 70.92 | 99.87 | 95.70 | 79.00 |
| | K-means | Median | 98.50 | 91.17 | 57.97 | 99.46 | 94.04 | 73.63 | 99.89 | 95.60 | 81.12 |
| GDINO Awa Low | Hierarchical | Mean | 98.60 | 90.61 | 50.31 | 99.44 | 93.87 | 69.95 | 99.87 | 95.83 | 78.25 |
| | Hierarchical | Median | 98.60 | 90.81 | 57.84 | 99.44 | 94.15 | 73.92 | 99.87 | 95.62 | 81.17 |
| | K-means | Mean | 98.52 | 90.79 | 63.00 | 99.39 | 93.94 | 77.13 | 99.90 | 95.57 | 82.22 |
| | K-means | Median | 98.57 | 90.91 | 65.43 | 99.41 | 94.17 | 77.70 | 99.87 | 95.67 | 83.09 |
| GDINO Partimagenet | Hierarchical | Mean | 98.44 | 90.99 | 53.57 | 99.41 | 94.17 | 71.35 | 99.80 | 95.57 | 79.01 |
| | Hierarchical | Median | 98.47 | 91.30 | 59.30 | 99.39 | 94.12 | 74.30 | 99.80 | 95.54 | 81.14 |
| | K-means | Mean | 98.47 | 91.19 | 53.92 | 99.39 | 94.10 | 71.54 | 99.82 | 95.57 | 78.79 |
| | K-means | Median | 98.47 | 91.27 | 59.63 | 99.44 | 94.17 | 74.70 | 99.82 | 95.62 | 81.52 |
| GDINO Partimagenet Low | Hierarchical | Mean | 98.42 | 90.63 | 52.42 | 99.46 | 94.20 | 70.66 | 99.85 | 95.60 | 78.20 |
| | Hierarchical | Median | 98.55 | 90.76 | 58.99 | 99.46 | 94.25 | 74.44 | 99.82 | 95.55 | 81.14 |
| | K-means | Mean | 98.42 | 90.68 | 51.97 | 99.49 | 94.30 | 70.56 | 99.84 | 95.62 | 78.12 |
| | K-means | Median | 98.52 | 90.99 | 59.39 | 99.52 | 94.17 | 74.20 | 99.85 | 95.83 | 81.22 |
| GDINO Pascal | Hierarchical | Mean | 98.60 | 90.60 | 51.00 | 99.44 | 93.86 | 70.02 | 99.87 | 95.82 | 77.99 |
| | Hierarchical | Median | 98.60 | 90.81 | 58.77 | 99.44 | 94.15 | 73.66 | 99.87 | 95.62 | 80.74 |
| | K-means | Mean | 98.52 | 90.79 | 50.95 | 99.39 | 93.94 | 70.26 | 99.90 | 95.57 | 77.93 |
| | K-means | Median | 98.57 | 90.91 | 58.80 | 99.41 | 93.17 | 73.73 | 99.87 | 95.67 | 81.00 |
| GDINO Pascal Low | Hierarchical | Mean | 98.62 | 90.43 | 50.17 | 99.46 | 93.84 | 69.66 | 99.87 | 95.90 | 77.61 |
| | Hierarchical | Median | 98.62 | 90.84 | 58.37 | 99.49 | 94.10 | 73.97 | 99.85 | 95.75 | 81.03 |
| | K-means | Mean | 98.60 | 90.58 | 50.48 | 99.49 | 93.99 | 69.64 | 99.85 | 95.80 | 77.51 |
| | K-means | Median | 98.62 | 90.89 | 58.25 | 99.52 | 93.89 | 73.78 | 99.85 | 95.75 | 81.15 |
| GDINO Sun | Hierarchical | Mean | 98.11 | 82.77 | 58.53 | 99.13 | 88.52 | 73.71 | 99.62 | 93.33 | 80.91 |
| | Hierarchical | Median | 98.09 | 83.10 | 58.85 | 99.16 | 88.90 | 73.70 | 99.62 | 93.40 | 80.95 |
| | K-means | Mean | 98.09 | 83.13 | 58.72 | 99.13 | 88.62 | 73.66 | 99.57 | 93.54 | 80.84 |
| | K-means | Median | 98.16 | 82.97 | 58.85 | 99.18 | 88.78 | 73.77 | 99.62 | 93.54 | 81.00 |
| GDINO Sun Low | Hierarchical | Mean | 98.50 | 90.94 | 54.49 | 99.49 | 93.71 | 72.39 | 99.82 | 95.69 | 79.43 |
| | Hierarchical | Median | 98.60 | 90.94 | 58.94 | 99.47 | 93.79 | 74.42 | 99.82 | 95.88 | 81.36 |
| | K-means | Mean | 98.55 | 90.91 | 54.80 | 99.49 | 93.43 | 72.61 | 99.80 | 95.75 | 80.00 |
| | K-means | Median | 98.60 | 90.94 | 59.00 | 99.47 | 94.46 | 74.54 | 99.80 | 95.88 | 81.51 |

## D.5. Number of clusters

As described in Section 3, we cluster the concept proposal set $\mathcal{S}$ into $k$ clusters. The choice of $k$ determines the granularity of the concepts used in the DCBM method. To analyze this, we test various values of $k$ within the K-Means algorithm, as shown in Table 16, presenting an ablation study of $k$. We do not impose the 50-images-per-class limitation in these experiments due to the generally smaller number of images available per dataset and class. As the table illustrates, increasing the number of clusters leads to a corresponding improvement in accuracy scores, highlighting the benefit of finer-grained concept representations.

*Table 16.* **Number of clusters.** Here we ablate the impact of different values of $k$ for K-Means within DCBM.

| Model | Clusters | CLIP ResNet-50 | | | CLIP ViT-B/16 | | | CLIP ViT-L/14 | | |
|---|---|---|---|---|---|---|---|---|---|---|
| | | ImageNette | ImageWoof | CUB | ImageNette | ImageWoof | CUB | ImageNette | ImageWoof | CUB |
| SAM | 128 | 97.73 | 82.29 | 37.40 | 99.13 | 89.41 | 59.82 | 99.64 | 93.05 | 71.35 |
| | 256 | 98.27 | 86.77 | 46.58 | 99.31 | 92.08 | 68.61 | 99.80 | 95.06 | 76.61 |
| | 512 | 98.60 | 89.57 | 53.38 | 99.46 | 93.51 | 72.11 | 99.82 | 95.65 | 78.94 |
| | 1024 | 98.50 | 90.81 | 57.62 | 99.54 | 93.79 | 73.44 | 99.80 | 97.78 | 79.58 |
| | 2048 | – | – | 59.16 | – | – | 73.85 | – | – | 80.44 |
| SAM2 | 128 | 98.22 | 87.02 | 44.68 | 99.36 | 92.19 | 65.29 | 99.80 | 95.50 | 75.56 |
| | 256 | 98.42 | 89.69 | 51.28 | 99.46 | 93.84 | 71.92 | 99.75 | 95.60 | 59.38 |
| | 512 | 98.62 | 91.14 | 57.21 | 99.35 | 94.35 | 74.70 | 99.80 | 95.78 | 81.55 |
| | 1024 | 98.78 | 91.68 | 61.44 | 99.57 | 94.30 | 75.49 | 99.85 | 95.80 | 82.21 |
| | 2048 | – | – | 62.98 | – | – | 75.80 | – | – | 82.27 |
| DETR | 128 | 98.42 | 89.62 | 52.07 | 99.23 | 93.66 | 70.62 | 99.75 | 95.42 | 78.03 |
| | 256 | 98.83 | 90.81 | 59.20 | 99.42 | 94.30 | 74.59 | 99.77 | 95.62 | 80.62 |
| | 512 | 98.58 | 91.81 | 63.48 | 99.47 | 94.50 | 76.41 | 99.77 | 95.67 | 82.24 |
| | 1024 | 98.73 | 92.03 | 61.18 | 99.54 | 94.58 | 76.91 | 99.80 | 95.78 | 82.52 |
| | 2048 | – | – | 64.27 | – | – | 76.67 | – | – | 82.65 |
| MaskRCNN | 128 | 98.37 | 88.67 | 54.95 | 99.44 | 93.41 | 74.18 | 99.80 | 95.34 | 80.07 |
| | 256 | 98.56 | 90.38 | 61.86 | 99.41 | 94.17 | 76.79 | 99.80 | 95.70 | 82.31 |
| | 512 | 98.57 | 91.48 | 62.50 | 99.52 | 94.45 | 77.84 | 99.82 | 95.57 | 83.28 |
| | 1024 | 98.68 | 91.89 | 65.46 | 99.59 | 94.60 | 77.61 | 99.82 | 95.65 | 83.32 |
| | 2048 | 98.80 | 92.11 | 65.71 | 99.62 | 94.53 | 77.39 | 99.87 | 95.60 | 83.34 |
| GDINO Awa | 128 | 98.19 | 85.52 | 40.94 | 99.34 | 91.19 | 64.95 | 99.64 | 94.02 | 74.20 |
| | 256 | 98.19 | 89.08 | 49.74 | 99.39 | 93.48 | 70.66 | 99.82 | 95.39 | 79.25 |
| | 512 | 98.42 | 90.58 | 55.13 | 99.34 | 93.94 | 73.00 | 99.82 | 95.34 | 80.31 |
| | 1024 | – | 91.17 | 58.60 | – | 94.04 | 73.63 | – | 95.72 | 81.20 |
| | 2048 | – | – | 60.18 | – | – | 74.68 | – | – | 81.53 |
| GDINO Awa Low | 128 | 98.19 | 84.37 | 39.23 | 99.34 | 90.81 | 62.43 | 99.72 | 94.40 | 72.37 |
| | 256 | 98.19 | 88.19 | 47.76 | 99.29 | 92.90 | 69.55 | 99.82 | 95.01 | 78.48 |
| | 512 | 98.39 | 90.18 | 54.06 | 99.31 | 93.82 | 72.44 | 99.87 | 95.62 | 80.67 |
| | 1024 | 98.50 | 91.22 | 57.82 | 99.47 | 94.02 | 74.00 | 99.90 | 95.72 | 81.41 |
| | 2048 | – | – | 59.58 | – | – | 74.32 | – | – | 81.62 |
| GDINO Partimagenet | 128 | 97.91 | 86.97 | 41.91 | 99.36 | 91.09 | 66.32 | 99.75 | 94.42 | 75.20 |
| | 256 | 98.29 | 89.64 | 50.50 | 99.31 | 93.33 | 71.33 | 99.70 | 95.19 | 79.03 |
| | 512 | 98.39 | 90.38 | 56.42 | 99.43 | 93.89 | 73.80 | 99.82 | 95.47 | 80.43 |
| | 1024 | 98.52 | 91.20 | 59.63 | 99.43 | 94.14 | 74.70 | 99.80 | 95.62 | 81.51 |
| | 2048 | – | – | 60.36 | – | – | 74.88 | – | – | 81.91 |
| GDINO Partimagenet Low | 128 | 97.99 | 84.55 | 43.17 | 99.36 | 91.02 | 65.98 | 99.75 | 94.25 | 74.87 |
| | 256 | 98.17 | 88.34 | 50.64 | 99.39 | 93.33 | 70.52 | 99.77 | 95.39 | 78.65 |
| | 512 | 98.45 | 90.25 | 56.08 | 99.47 | 93.63 | 73.25 | 99.82 | 95.62 | 80.41 |
| | 1024 | 98.52 | 90.99 | 59.39 | 99.49 | 94.17 | 74.20 | 99.85 | 95.60 | 81.22 |
| | 2048 | – | – | 60.04 | – | – | 74.59 | – | – | 81.72 |
| GDINO Pascal | 128 | 98.04 | 84.23 | 39.56 | 99.16 | 91.52 | 63.43 | 99.67 | 93.82 | 73.77 |
| | 256 | 98.45 | 88.70 | 48.33 | 99.29 | 93.13 | 69.04 | 99.82 | 95.24 | 77.91 |
| | 512 | 98.47 | 90.48 | 54.35 | 99.41 | 93.82 | 72.54 | 99.80 | 95.39 | 80.29 |
| | 1024 | – | 91.02 | 58.27 | – | 93.84 | 73.71 | – | 95.78 | 81.00 |
| | 2048 | – | – | 59.37 | – | – | 74.20 | – | – | 81.38 |
| GDINO Pascal Low | 128 | 98.27 | 84.58 | 40.75 | 99.29 | 90.43 | 65.15 | 99.77 | 93.66 | 73.16 |
| | 256 | 98.24 | 88.55 | 49.50 | 99.41 | 93.00 | 70.21 | 99.82 | 95.14 | 78.49 |
| | 512 | 98.50 | 90.25 | 54.71 | 99.43 | 93.64 | 72.47 | 99.85 | 95.32 | 80.27 |
| | 1024 | 98.60 | 90.91 | 58.25 | 99.52 | 94.04 | 73.89 | 99.85 | 95.67 | 81.15 |
| | 2048 | – | – | 59.84 | – | – | 74.04 | – | – | 81.69 |
| GDINO Sun | 128 | 98.11 | 82.99 | 43.04 | 99.18 | 88.85 | 66.76 | 99.62 | 93.56 | 74.94 |
| | 256 | – | – | 52.64 | – | – | 71.26 | – | – | 78.81 |
| | 512 | – | – | 56.92 | – | – | 73.52 | – | – | 80.55 |
| | 1024 | – | – | 58.66 | – | – | 73.75 | – | – | 81.00 |
| GDINO Sun Low | 128 | 97.99 | 84.81 | 42.37 | 99.23 | 90.91 | 66.17 | 99.72 | 93.92 | 75.06 |
| | 256 | 98.29 | 88.34 | 49.64 | 99.42 | 92.84 | 71.73 | 99.75 | 95.09 | 79.08 |
| | 512 | 98.57 | 90.22 | 55.87 | 99.44 | 93.69 | 73.30 | 99.82 | 95.55 | 80.29 |
| | 1024 | 98.60 | 90.99 | 58.94 | 99.49 | 93.48 | 74.59 | 99.80 | 95.85 | 81.34 |
| | 2048 | – | – | 60.03 | – | – | 75.08 | – | – | 81.58 |

## D.6. Sparsity parameter $\lambda$ and learning rate

*Table 17.* **Part 1: learning rates and sparsity.** Performance across CLIP ResNet-50, ViT-B/16, and ViT-L/14 on the ImageNette, ImageWoof, and CUB datasets with varying learning rates and sparsity parameters for segmentation models.

| Model | Configuration | | CLIP ResNet-50 | | | CLIP ViT-B/16 | | | CLIP ViT-L/14 | | |
|---|---|---|---|---|---|---|---|---|---|---|---|
| | Learning Rate | Sparsity | ImageNette | ImageWoof | CUB | ImageNette | ImageWoof | CUB | ImageNette | ImageWoof | CUB |
| SAM | 1e-4 | 1e-4 | 98.50 | 90.81 | 57.71 | 99.72 | 93.74 | 73.14 | 99.87 | 95.67 | 79.73 |
| | 1e-3 | 1e-4 | 98.24 | 89.41 | 43.23 | 99.31 | 92.54 | 66.38 | 99.75 | 95.19 | 74.68 |
| | 1e-2 | 1e-4 | 97.68 | 80.73 | 6.18 | 99.08 | 86.03 | 12.01 | 99.77 | 91.62 | 14.58 |
| | 1e-4 | 1e-3 | 98.57 | 91.09 | 54.59 | 99.44 | 93.87 | 69.38 | 99.77 | 95.72 | 76.46 |
| | 1e-4 | 1e-2 | 98.45 | 89.49 | 47.38 | 99.44 | 93.15 | 63.05 | 99.77 | 95.42 | 69.62 |
| | 1e-3 | 1e-3 | 98.14 | 89.18 | 38.06 | 99.29 | 92.52 | 58.50 | 99.72 | 94.99 | 68.43 |
| | 1e-2 | 1e-2 | 97.25 | 68.41 | 3.37 | 98.88 | 82.44 | 6.77 | 99.54 | 90.22 | 10.44 |
| SAM2 | 1e-4 | 1e-4 | 98.75 | 91.60 | 61.44 | 99.51 | 94.32 | 75.49 | 99.90 | 95.75 | 82.21 |
| | 1e-3 | 1e-4 | 98.55 | 90.66 | 49.86 | 99.59 | 93.84 | 70.33 | 99.80 | 95.72 | 77.55 |
| | 1e-2 | 1e-4 | 97.76 | 85.01 | 6.99 | 99.44 | 90.38 | 14.07 | 99.69 | 93.94 | 21.45 |
| | 1e-4 | 1e-3 | 98.78 | 91.50 | 58.15 | 99.49 | 94.02 | 71.97 | 99.90 | 95.39 | 78.91 |
| | 1e-4 | 1e-2 | 98.62 | 90.02 | 51.19 | 99.49 | 93.00 | 64.81 | 99.85 | 96.60 | 72.04 |
| | 1e-3 | 1e-3 | 98.50 | 90.23 | 43.94 | 99.57 | 93.71 | 62.05 | 99.80 | 95.57 | 70.54 |
| | 1e-2 | 1e-2 | 97.22 | 77.07 | 4.38 | 99.06 | 83.66 | 10.30 | 99.70 | 91.19 | 15.74 |
| DETR | 1e-4 | 1e-4 | 98.75 | 92.03 | 65.20 | 99.52 | 94.58 | 76.91 | 99.79 | 95.80 | 82.64 |
| | 1e-3 | 1e-4 | 98.55 | 91.11 | 56.32 | 99.46 | 94.19 | 72.92 | 99.77 | 95.72 | 78.63 |
| | 1e-2 | 1e-4 | 97.96 | 84.55 | 10.80 | 99.31 | 91.70 | 16.95 | 99.69 | 94.53 | 17.69 |
| | 1e-4 | 1e-3 | 91.75 | 91.75 | 60.04 | 99.49 | 94.25 | 72.66 | 99.80 | 95.75 | 79.50 |
| | 1e-4 | 1e-2 | 90.40 | 90.40 | 54.16 | 99.49 | 93.31 | 66.95 | 99.77 | 95.27 | 72.97 |
| | 1e-3 | 1e-3 | 98.62 | 90.48 | 50.41 | 99.39 | 93.89 | 75.74 | 99.77 | 95.55 | 72.71 |
| | 1e-2 | 1e-2 | 97.27 | 83.69 | 6.51 | 98.77 | 86.92 | 10.08 | 99.44 | 92.44 | 13.36 |
| MaskRCNN | 1e-4 | 1e-4 | 98.73 | 91.93 | 65.46 | 99.61 | 94.60 | 77.61 | 99.85 | 95.72 | 83.33 |
| | 1e-3 | 1e-4 | 98.47 | 90.96 | 58.72 | 99.51 | 94.25 | 74.63 | 99.77 | 95.65 | 81.08 |
| | 1e-2 | 1e-4 | 97.91 | 85.39 | 13.69 | 99.26 | 91.70 | 31.05 | 99.69 | 95.43 | 50.38 |
| | 1e-4 | 1e-3 | 98.70 | 90.25 | 61.48 | 99.52 | 94.17 | 73.73 | 99.82 | 95.55 | 80.19 |
| | 1e-4 | 1e-2 | 98.57 | 89.92 | 55.75 | 99.46 | 93.26 | 68.52 | 99.85 | 92.19 | 75.80 |
| | 1e-3 | 1e-3 | 98.50 | 90.25 | 52.19 | 99.41 | 93.84 | 67.83 | 99.77 | 95.39 | 74.94 |
| | 1e-2 | 1e-2 | 97.22 | 78.01 | 8.70 | 98.75 | 84.96 | 16.88 | 99.57 | 92.19 | 28.55 |

In Table 17 and Table 18, we analyze the impact of different hyperparameters for sparsity ($\lambda$) in Equation (3) and the learning rate. To provide a more transparent overview, we present the results in two separate tables: Table 17 focuses on segmentation models, while Table 18 covers object detection models. All other configurations remain consistent with the settings reported earlier. From the results, the optimal configuration across the three datasets and various concept creation modules is a learning rate of $1e-4$ combined with a sparsity of $1e-4$. Although specific configurations demonstrate improved performance with alternative values, we aim to identify a baseline configuration that performs robustly across diverse scenarios.

*Table 18.* **Part 2: learning rates and sparsity.** Performance across CLIP ResNet-50, ViT-B/16, and ViT-L/14 on the ImageNette, ImageWoof, and CUB datasets with varying learning rates and sparsity parameters for object detection models.

| Model | Configuration | | CLIP ResNet-50 | | | CLIP ViT-B/16 | | | CLIP ViT-L/14 | | |
| --- | --- | --- | --- | --- | --- | --- | --- | --- | --- | --- | --- |
| | Learning Rate | Sparsity | ImageNette | ImageWoof | CUB | ImageNette | ImageWoof | CUB | ImageNette | ImageWoof | CUB |
| GDINO Awa | 1e-4 | 1e-4 | 98.49 | 91.27 | 58.44 | 99.46 | 94.94 | 73.63 | 99.82 | 95.67 | 81.20 |
| | 1e-3 | 1e-4 | 98.29 | 89.31 | 46.88 | 99.44 | 93.36 | 68.23 | 99.77 | 95.75 | 76.34 |
| | 1e-2 | 1e-4 | 97.53 | 81.65 | 7.37 | 99.26 | 87.25 | 14.72 | 99.64 | 93.33 | 17.67 |
| | 1e-4 | 1e-3 | 98.62 | 91.14 | 55.21 | 99.41 | 93.64 | 69.68 | 99.80 | 95.75 | 77.74 |
| | 1e-4 | 1e-2 | 98.44 | 89.28 | 48.43 | 99.44 | 93.13 | 63.41 | 99.77 | 94.33 | 71.63 |
| | 1e-3 | 1e-3 | 98.32 | 89.31 | 41.04 | 99.39 | 93.08 | 61.18 | 99.77 | 95.50 | 70.31 |
| | 1e-2 | 1e-2 | 96.56 | 70.71 | 4.25 | 98.93 | 79.13 | 7.46 | 99.49 | 88.83 | 11.66 |
| GDINO Awa Low | 1e-4 | 1e-4 | 98.55 | 91.22 | 57.97 | 99.47 | 94.02 | 73.63 | 99.89 | 95.72 | 81.11 |
| | 1e-3 | 1e-4 | 98.32 | 89.44 | 46.29 | 99.36 | 93.31 | 68.19 | 99.82 | 95.80 | 76.61 |
| | 1e-2 | 1e-4 | 97.61 | 80.63 | 7.13 | 99.13 | 88.09 | 13.46 | 99.67 | 93.38 | 17.74 |
| | 1e-4 | 1e-3 | 98.60 | 91.27 | 55.04 | 99.44 | 93.74 | 69.45 | 99.92 | 95.52 | 77.65 |
| | 1e-4 | 1e-2 | 98.42 | 88.95 | 48.55 | 99.36 | 92.72 | 63.39 | 99.89 | 95.04 | 70.59 |
| | 1e-3 | 1e-3 | 98.34 | 89.13 | 40.01 | 99.36 | 93.20 | 99.36 | 99.85 | 95.52 | 69.49 |
| | 1e-2 | 1e-2 | 96.51 | 69.89 | 4.16 | 98.95 | 90.10 | 6.97 | 99.67 | 89.03 | 10.99 |
| GDINO Partimagenet | 1e-4 | 1e-4 | 98.52 | 91.14 | 59.32 | 99.43 | 94.17 | 74.70 | 99.82 | 95.60 | 81.52 |
| | 1e-3 | 1e-4 | 98.06 | 89.92 | 49.72 | 99.31 | 93.54 | 69.19 | 99.69 | 95.67 | 77.56 |
| | 1e-2 | 1e-4 | 97.04 | 81.75 | 8.92 | 99.08 | 89.28 | 14.88 | 99.36 | 93.20 | 20.78 |
| | 1e-4 | 1e-3 | 98.47 | 91.30 | 56.33 | 99.45 | 93.84 | 70.88 | 99.80 | 95.42 | 78.39 |
| | 1e-4 | 1e-2 | 98.32 | 89.39 | 49.84 | 99.36 | 92.62 | 64.36 | 99.72 | 94.63 | 72.30 |
| | 1e-3 | 1e-3 | 98.04 | 89.39 | 43.27 | 99.34 | 93.64 | 62.05 | 99.71 | 95.27 | 71.73 |
| | 1e-2 | 1e-2 | 95.95 | 70.30 | 4.49 | 98.98 | 80.99 | 5.53 | – | – | 13.39 |
| GDINO Partimagenet Low | 1e-4 | 1e-4 | 98.52 | 90.99 | 59.39 | 99.49 | 94.22 | 74.20 | 99.85 | 95.78 | 81.22 |
| | 1e-3 | 1e-4 | 98.34 | 89.92 | 48.07 | 99.38 | 93.61 | 69.16 | 99.80 | 95.67 | 77.29 |
| | 1e-2 | 1e-4 | 97.35 | 81.01 | 7.73 | 99.18 | 88.78 | 15.59 | 99.64 | 93.10 | 17.93 |
| | 1e-4 | 1e-3 | 98.77 | 91.09 | 56.14 | 99.41 | 93.79 | 70.19 | 99.82 | 95.44 | 78.05 |
| | 1e-4 | 1e-2 | 98.39 | 89.16 | 49.07 | 99.36 | 92.92 | 64.38 | 99.80 | 94.88 | 74.07 |
| | 1e-3 | 1e-3 | 98.27 | 89.92 | 42.03 | 99.41 | 93.33 | 61.70 | 99.82 | 95.67 | 77.29 |
| | 1e-2 | 1e-2 | 96.36 | 72.82 | 4.26 | 99.21 | 80.55 | 7.82 | 99.72 | 89.82 | 12.02 |
| GDINO Pascal | 1e-4 | 1e-4 | 98.42 | 91.09 | 58.80 | 99.41 | 93.76 | 73.71 | 99.82 | 95.75 | 80.77 |
| | 1e-3 | 1e-4 | 98.04 | 89.67 | 46.70 | 99.18 | 93.03 | 68.47 | 99.67 | 95.78 | 76.39 |
| | 1e-2 | 1e-4 | 96.82 | 80.76 | 7.49 | 98.50 | 87.12 | 12.84 | 99.29 | 92.95 | 18.31 |
| | 1e-4 | 1e-3 | 98.50 | 91.32 | 55.71 | 99.46 | 93.66 | 69.74 | 99.80 | 95.09 | 77.52 |
| | 1e-4 | 1e-2 | 98.37 | 88.98 | 48.58 | 99.31 | 93.00 | 63.62 | 99.77 | 94.45 | 71.33 |
| | 1e-3 | 1e-3 | 98.14 | 89.03 | 42.47 | 99.21 | 92.85 | 60.54 | 99.69 | 95.44 | 69.42 |
| | 1e-2 | 1e-2 | 96.64 | 69.41 | 4.38 | 98.60 | 82.90 | 7.66 | 99.39 | 88.72 | 10.56 |
| GDINO Pascal Low | 1e-4 | 1e-4 | 98.88 | 90.91 | 58.25 | 99.46 | 94.20 | 73.78 | 99.85 | 95.75 | 81.89 |
| | 1e-3 | 1e-4 | 98.40 | 89.69 | 46.59 | 99.46 | 93.56 | 67.97 | 99.77 | 95.72 | 76.73 |
| | 1e-2 | 1e-4 | 97.55 | 81.88 | 8.84 | 99.11 | 89.36 | 15.03 | 99.64 | 93.66 | 17.60 |
| | 1e-4 | 1e-3 | 98.57 | 91.11 | 55.32 | 99.52 | 93.79 | 69.88 | 99.85 | 95.44 | 77.89 |
| | 1e-4 | 1e-2 | 98.42 | 89.69 | 48.71 | 99.34 | 93.05 | 63.31 | 99.85 | 94.50 | 71.19 |
| | 1e-3 | 1e-3 | 98.57 | 89.20 | 40.85 | 99.41 | 93.05 | 60.41 | 99.77 | 95.29 | 70.83 |
| | 1e-2 | 1e-2 | 98.42 | 70.20 | 4.11 | 98.90 | 81.14 | 7.75 | 99.72 | 88.47 | 10.75 |
| GDINO Sun | 1e-4 | 1e-4 | 98.09 | 82.97 | 58.66 | 99.18 | 88.62 | 73.66 | 99.59 | 93.63 | 81.03 |
| | 1e-3 | 1e-4 | 97.81 | 81.47 | 48.38 | 98.96 | 88.14 | 69.05 | 99.52 | 92.90 | 77.22 |
| | 1e-2 | 1e-4 | 95.46 | 70.04 | 8.42 | 98.04 | 77.65 | 13.94 | 98.90 | 86.84 | 19.28 |
| | 1e-4 | 1e-3 | 98.39 | 87.10 | 55.42 | 99.24 | 90.58 | 70.26 | 99.67 | 94.38 | 77.72 |
| | 1e-4 | 1e-2 | 98.24 | 87.02 | 49.31 | 99.24 | 90.26 | 63.29 | 99.62 | 93.74 | 70.64 |
| | 1e-3 | 1e-3 | 97.94 | 84.45 | 42.22 | 99.01 | 89.06 | 61.53 | 99.59 | 93.33 | 71.28 |
| | 1e-2 | 1e-2 | 95.75 | 68.49 | 4.99 | 98.17 | 76.43 | 7.92 | 99.03 | 86.82 | 12.36 |
| GDINO Sun Low | 1e-4 | 1e-4 | 98.57 | 90.94 | 58.63 | 99.49 | 93.46 | 74.59 | 99.82 | 95.88 | 81.33 |
| | 1e-3 | 1e-4 | 98.37 | 89.62 | 48.29 | 99.49 | 92.14 | 69.54 | 99.77 | 95.06 | 77.55 |
| | 1e-2 | 1e-4 | 97.58 | 80.71 | 9.04 | 99.18 | 83.10 | 15.78 | 99.69 | 90.91 | 18.97 |
| | 1e-4 | 1e-3 | 98.57 | 91.17 | 55.85 | 99.44 | 93.67 | 70.42 | 99.79 | 95.55 | 77.84 |
| | 1e-4 | 1e-2 | 98.42 | 88.78 | 49.57 | 99.41 | 92.77 | 64.64 | 99.80 | 95.09 | 70.90 |
| | 1e-3 | 1e-3 | 98.32 | 89.03 | 42.58 | 99.36 | 92.21 | 61.98 | 99.77 | 94.99 | 71.71 |
| | 1e-2 | 1e-2 | 96.33 | 69.71 | 4.76 | 98.96 | 77.60 | 8.37 | 99.67 | 87.78 | 11.91 |

## D.7. Number of images per class

DCBM does not require using all images in a dataset to generate meaningful concepts. In this section, we evaluate the impact of varying the number of images per class (5, 10, 25, and 50) on the resulting accuracy using the standard configurations from the ablation study. The accuracy scores are presented in Table 19. Increasing the number of images per class most often leads to improved accuracy. However, the improvements become marginal beyond 25 images, with the difference between 25 and 50 being minimal. Consequently, we limited our investigation to 50 images per class to balance accuracy with computational efficiency, as a higher number of images requires more segmentation and object detection, significantly increasing computational costs with diminishing returns.

*Table 19.* **Images per class.** Performance comparison across models using k-means clustering (1024 clusters) with varying numbers of images per class.

| Model | Images/Cls | CLIP ResNet-50 | | | CLIP ViT-B/16 | | | CLIP ViT-L/14 | | |
|---|---|---|---|---|---|---|---|---|---|---|
| | | ImageNette | ImageWoof | CUB | ImageNette | ImageWoof | CUB | ImageNette | ImageWoof | CUB |
| SAM | 5 | 98.47 | 91.14 | 58.78 | 99.52 | 93.94 | 80.26 | 99.80 | 95.52 | 80.26 |
| | 10 | 98.73 | 91.27 | 57.82 | 99.54 | 93.94 | 74.13 | 99.80 | 95.50 | 80.03 |
| | 25 | 98.57 | 91.07 | 57.53 | 99.54 | 94.07 | 73.35 | 99.85 | 95.50 | 79.82 |
| | 50 | 98.50 | 90.81 | 57.71 | 99.72 | 93.74 | 73.14 | 99.87 | 95.67 | 79.73 |
| SAM2 | 5 | 98.39 | – | – | 99.46 | – | – | 99.82 | – | – |
| | 10 | 98.70 | 86.49 | 63.86 | 99.36 | 92.21 | 76.60 | 99.82 | 94.50 | 82.57 |
| | 25 | 98.68 | 89.64 | 63.63 | 99.54 | 93.61 | 76.48 | 99.85 | 95.32 | 82.45 |
| | 50 | 98.75 | 91.93 | 61.32 | 99.52 | 94.45 | 75.49 | 99.87 | 95.80 | 82.44 |
| DETR | 5 | 98.32 | 88.88 | 62.75 | 99.29 | 93.79 | 76.29 | 99.77 | 95.37 | 82.38 |
| | 10 | 98.70 | 91.14 | 63.88 | 99.59 | 94.35 | 76.84 | 99.80 | 95.72 | 82.60 |
| | 25 | 98.62 | 91.80 | 63.89 | 99.41 | 94.58 | 76.82 | 99.80 | 95.67 | 82.76 |
| | 50 | 98.75 | 92.03 | 65.20 | 99.52 | 94.58 | 76.91 | 99.79 | 95.80 | 82.64 |
| MaskRCNN | 5 | 98.47 | 90.48 | 65.93 | 99.46 | 94.12 | 83.47 | 99.82 | 95.78 | 77.77 |
| | 10 | 98.65 | 91.27 | 65.60 | 99.57 | 94.32 | 77.70 | 99.77 | 95.75 | 83.26 |
| | 25 | 98.68 | 91.68 | 65.27 | 99.54 | 94.76 | 77.84 | 99.87 | 95.67 | 83.40 |
| | 50 | 98.73 | 91.86 | 65.46 | 99.60 | 94.60 | 77.70 | 99.85 | 95.65 | 83.33 |
| GDINO Awa | 5 | – | 84.78 | 59.99 | – | 90.66 | 74.42 | – | 93.71 | 81.69 |
| | 10 | 98.34 | 88.88 | 59.51 | 99.28 | 93.00 | 74.58 | 99.75 | 95.29 | 81.57 |
| | 25 | 98.42 | 90.56 | 58.84 | 99.29 | 93.82 | 74.47 | 99.82 | 95.55 | 81.26 |
| | 50 | 98.50 | 91.17 | 57.97 | 99.46 | 94.04 | 73.63 | 99.89 | 95.60 | 81.12 |
| GDINO Awa Low | 5 | 98.44 | 88.90 | 59.65 | 99.36 | 93.10 | 74.44 | 99.80 | 95.39 | 81.86 |
| | 10 | 98.57 | 90.56 | 58.84 | 99.31 | 93.54 | 74.51 | 99.85 | 95.60 | 81.61 |
| | 25 | 98.57 | 90.99 | 57.59 | 99.46 | 94.02 | 73.51 | 99.85 | 95.67 | 81.27 |
| | 50 | 98.55 | 91.22 | 57.97 | 99.47 | 94.02 | 73.63 | 99.89 | 95.72 | 81.11 |
| GDINO Partimagenet | 5 | 97.99 | 86.33 | 60.18 | 99.54 | 92.16 | 74.84 | 98.98 | 94.94 | 81.95 |
| | 10 | 98.32 | 89.44 | 60.17 | 99.72 | 93.26 | 74.75 | 99.13 | 95.50 | 81.81 |
| | 25 | 98.32 | 90.63 | 59.41 | 99.80 | 94.15 | 73.85 | 99.31 | 95.60 | 81.10 |
| | 50 | 98.52 | 91.14 | 59.32 | 99.43 | 94.17 | 74.70 | 99.82 | 95.60 | 81.52 |
| GDINO Partimagenet Low | 5 | 98.32 | 89.16 | 60.27 | 99.31 | 93.33 | 74.42 | 99.82 | 95.55 | 81.53 |
| | 10 | 98.40 | 90.58 | 59.82 | 99.36 | 93.92 | 74.44 | 99.80 | 95.83 | 81.46 |
| | 25 | 99.37 | 91.09 | 58.94 | 99.47 | 94.22 | 74.47 | 99.80 | 95.65 | 81.50 |
| | 50 | 98.52 | 90.99 | 59.39 | 99.49 | 94.22 | 74.20 | 99.85 | 95.78 | 81.22 |
| GDINO Pascal | 5 | – | 85.42 | 59.96 | – | 91.45 | 74.63 | – | 94.66 | 81.53 |
| | 10 | 98.99 | 88.31 | 59.51 | 99.08 | 93.10 | 74.32 | 99.69 | 95.44 | 81.15 |
| | 25 | 98.34 | 90.99 | 58.39 | 99.39 | 93.97 | 74.01 | 99.82 | 95.55 | 80.82 |
| | 50 | 98.57 | 90.91 | 58.80 | 99.41 | 93.17 | 73.73 | 99.87 | 95.67 | 81.00 |
| GDINO Pascal Low | 5 | 98.17 | 89.03 | 59.23 | 99.24 | 92.98 | 74.18 | 99.72 | 95.44 | 81.27 |
| | 10 | 98.32 | 90.68 | 59.42 | 99.36 | 93.59 | 74.56 | 99.80 | 95.70 | 81.31 |
| | 25 | 98.57 | 90.99 | 58.63 | 99.52 | 94.10 | 73.77 | 99.80 | 95.75 | 81.15 |
| | 50 | 98.62 | 90.89 | 58.25 | 99.52 | 93.89 | 73.78 | 99.85 | 95.75 | 81.15 |
| GDINO Sun | 5 | – | – | – | – | – | – | – | – | – |
| | 10 | – | – | – | – | – | – | – | – | – |
| | 25 | – | – | – | – | – | – | – | – | – |
| | 50 | 98.09 | 82.97 | 58.66 | 99.18 | 88.62 | 73.66 | 99.59 | 93.63 | 81.03 |
| GDINO Sun Low | 5 | 97.99 | 83.15 | 60.03 | 99.21 | 89.23 | 74.72 | 99.67 | 93.54 | 81.88 |
| | 10 | 98.32 | 87.48 | 60.01 | 99.34 | 92.42 | 74.89 | 99.72 | 95.22 | 81.62 |
| | 25 | 98.50 | 90.23 | 59.48 | 99.41 | 93.53 | 74.40 | 99.80 | 95.60 | 81.36 |
| | 50 | 98.60 | 90.94 | 59.00 | 99.47 | 94.46 | 74.54 | 99.80 | 95.88 | 81.51 |

# E. Implementation details

Our implementation is in Python and Pytorch, and our CBM implementation is based on (Rao et al., 2024; Yuksekgonul et al., 2023). For GradCAM (Selvaraju et al., 2019) calculations, we use the implementation by (Zakka, 2021), which we adjust to ViT's and concept instead of text matching based on Gildenblat *et al.*'s implementation (Gildenblat & contributors, 2021).

## E.1. Efficiency and runtimes

The creation of concept proposals is influenced by the number of images per class and the foundation model used to generate outputs from these images. Consequently, the computational effort required for concept creation is highly dependent on the input characteristics, similar to other approaches that involve training a sparse autoencoder for concept generation (Rao et al., 2024), performing non-negative matrix factorization, or utilizing a large language model (Yang et al., 2023). In the main paper, we provided rough estimations of the computation time required for concept creation within DCBM.

However, once the concepts are generated, we can directly compare the runtime per epoch between our framework and others on the same dataset. To illustrate this, we measured the runtime for a single epoch ImageNet using DCBM and DN-CBM, employing a 90/10 train-validation split and adhering to the standard hyperparameters specified in (Rao et al., 2024) and our work for DCBM. Both methods were evaluated on the same machine using ViT ResNet50 as the backbone. Our measurements reveal that a single epoch takes an average of 16 seconds for DN-CBM, while for DCBM, the average runtime per epoch is 9 seconds (with SAM2). This significant reduction in runtime for DCBM is attributed to fewer concepts, which reduces the input size to the linear model compared to DN-CBM.

## E.2. Concept size ablation

We remove the smallest and largest segments to ablate their effect on the CBM performance. To this end, we remove all concepts which are 1000 pixel or smaller (GT1000) and 1500 pixel or smaller(GT1500). Additionally, we remove all concepts that contain 200k or more pixels (LT200k).

*Table 20.* **Test accuracy across datasets and segmentation methods.** Comparison of test accuracy for different segmentation methods on CUB, ImageNet, and ClimateTV_animals datasets under various ground truth (GT) conditions.

| Dataset | Segmentation | Test Acc. | GT 1000 | GT 1500 | lt200k |
|---|---|---|---|---|---|
| CUB | SAM2 | 82.4 | 82.7 | 82.7 | 82.4 |
| CUB | GDINO_partimagenet | 81.8 | 81.8 | 81.4 | 81.7 |
| ImageNet | GDINO_partimagenet | 77.4 | 77.4 | 77.4 | 77.4 |
| ImageNet | SAM2 | 77.9 | 77.7 | 77.9 | 77.9 |
| ClimateTV_animals | SAM2 | 86.4 | 84.8 | 87.1 | 85.6 |
| ClimateTV_animals | GDINO_partimagenet | 83.3 | 82.6 | 87.1 | 85.6 |

# F. Quantitative results

## F.1. CLIP backbones

We present the accuracies of additional CLIP backbones, ResNet-50 and ViT B/16, in Table 21. We used the same hyperparameters as those initially selected for ViT L/14 for consistency. While ViT B/16 performs competitively, ResNet-50 lags behind other CBM techniques. We attribute this disparity to ResNet-50's limited capacity to capture complex semantic relationships compared to transformer-based models, better equipped to handle the nuanced contextual understanding required for this task.

*Table 21.* **Extended CBM benchmark.** Performance comparison across different CLIP versions on datasets used for ablation study. Hyperparameters correspond to the ones reported for ViT L14.

| Model | CLIP ResNet-50 | | | | | CLIP ViT-B/16 | | | | | CLIP ViT-L/14 | | | | |
|---|---|---|---|---|---|---|---|---|---|---|---|---|---|---|---|
| | IMN | Places | CUB | Cif10 | Cif100 | IMN | Places | CUB | Cif10 | Cif100 | IMN | Places | CUB | Cif10 | Cif100 |
| Linear Probe | 73.3* | 53.4* | 68.9 | 88.7* | 70.3* | 80.2* | 55.1* | 81.0 | 96.2* | 83.1* | 83.9* | 55.4 | 85.7 | 98.0* | 87.5* |
| Zero Shot | 59.6* | 37.9 | 46.1 | 75.6* | 41.6* | 68.6* | 39.5* | 55.0 | 91.6* | 68.7* | 75.3* | 40.0 | 62.2 | 96.2* | 77.9* |
| LF-CBM (Oikarinen et al., 2023) | 72.0* | 46.8 | 74.3* | 86.4* | 65.1* | 75.4 | 48.2 | 74.0 | 94.7 | 77.4* | – | 49.4 | 80.1 | 97.2 | 83.9 |
| LaBo (Yang et al., 2023) | 68.9* | – | – | 87.9* | 69.1* | 78.9* | – | – | 95.7* | 81.2* | 84.0* | – | – | 97.8* | 86.0* |
| CDM (Panousis et al., 2023) | 72.2* | 52.7* | 72.3* | 86.5* | 67.6* | 79.3* | 52.6* | 79.5* | 95.3* | 80.5* | 83.4* | 55.2* | – | 95.9 | 82.2 |
| DCLIP (Menon & Vondrick, 2023) | 59.6* | 37.9* | 49.0 | – | – | 68.0* | 40.3* | 57.8* | – | – | 75.0* | 40.5* | 63.5* | – | – |
| DN-CBM (Rao et al., 2024) | 72.9* | 53.5* | – | 87.6* | 67.5* | 79.5* | 55.1* | – | 96.0* | 82.1* | 83.6 | 55.6 | – | 98.1 | 86.0 |
| DCBM-SAM2 (Ours) | 58.7 | 48.0 | 61.4 | 84.5 | 61.8 | 70.4 | 50.6 | 75.3 | 95.2 | 79.4 | 77.9 | 52.1 | 81.8 | 97.7 | 85.4 |
| DCBM-GDINO (Ours) | 58.7 | 47.8 | 59.0 | 83.9 | 61.2 | 69.7 | 50.7 | 74.1 | 95.1 | 79.6 | 77.4 | 52.2 | 81.3 | 97.5 | 85.3 |
| DCBM-MASKRCNN (Ours) | 58.7 | 48.2 | 64.6 | 84.5 | 62.7 | 70.5 | 50.9 | 76.7 | 95.2 | 79.6 | 77.8 | 52.1 | 82.4 | 97.7 | 85.6 |

## F.2. Additional datasets

**AWA2** performance of DCBM is on par with the linear probe for all embeddings, as shown in Table 22.

| | ResNet-50 | ViT-B/16 | ViT-L/14 |
|---|---|---|---|
| Zero shot | 88.94 | 94.00 | 95.94 |
| Linear probe | 93.72 | 96.51 | 97.68 |
| DCBM | 93.13 | 96.43 | 97.71 |

*Table 22.* DCBM performance on AwA2 using GDINO (w/ partimagenet labels) as concept proposal method

**CelebA** performance of DCBM is superior to other methods reported by (Xu et al., 2024) as shown in Table 23. Given the label generation process, we believe that this is not the most suitable labeling scheme. We advocate for future research to devise more distinct, independent, and objective labels.

| CBM Model | Acc |
|---|---|
| CBM | 0.246 |
| ProbCBM | 0.299 |
| PCBM | 0.150 |
| CEM | 0.330 |
| ECBM | 0.343 |
| **DCBM w/ GDINO (ours)** | **0.354** |
| **DCBM w/ MaskRCNN (ours)** | **0.363** |
| **DCBM w/ SAM2 (ours)** | **0.356** |

*Table 23.* DCBM performance on CelebA using GDINO (w/ partimagenet labels) as concept proposal method (CLIP ViT-L/14)

**BotCL.** We conducted experiments on the first 200 classes of ImageNet, analogous to BotCL (Wang et al., 2023). DCBM achieved a test accuracy of 84.7% using CLIP ViT-L/14 and GroundingDINO (partimagenet). BotCL achieves 79.5% accuracy for this task. When comparing DCBM to BotCL, one has to bear in mind that the models have different backbones, limiting comparability.

## G. Qualitative results

In the following, we provide additional explanations in Figure 9, similar to those discussed in Section 4.3, focusing on the ablation datasets ImageWoof and ImageNette. These examples further highlight the effectiveness of DCBM on these specific datasets. Since both datasets consist of only ten classes each, the limited variety of concepts is likely a result of the reduced range of distinct concept proposals. This occurs because the datasets contain similar images representing closely related classes, leading to overlapping or redundant concept representations.

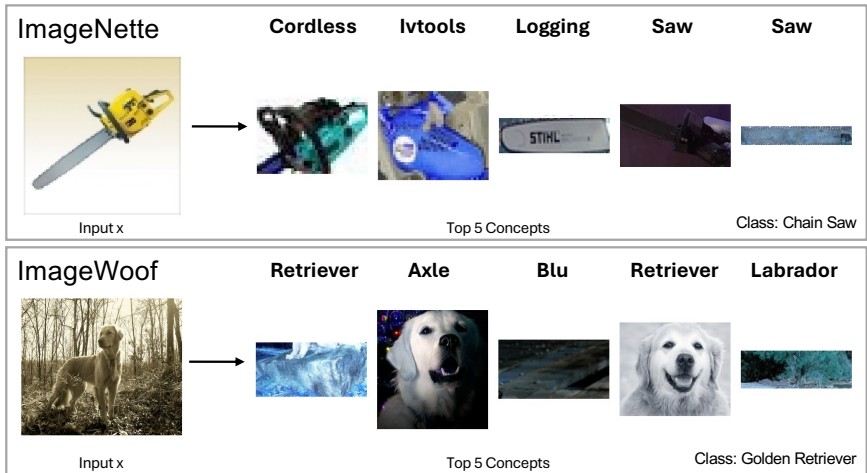

*Figure 9.* **DCBM ablations.** Our technique DCBM was applied to ImageWoof and ImageNette (SAM2 and ViT L/14).

### G.1. DCBM concept comparison

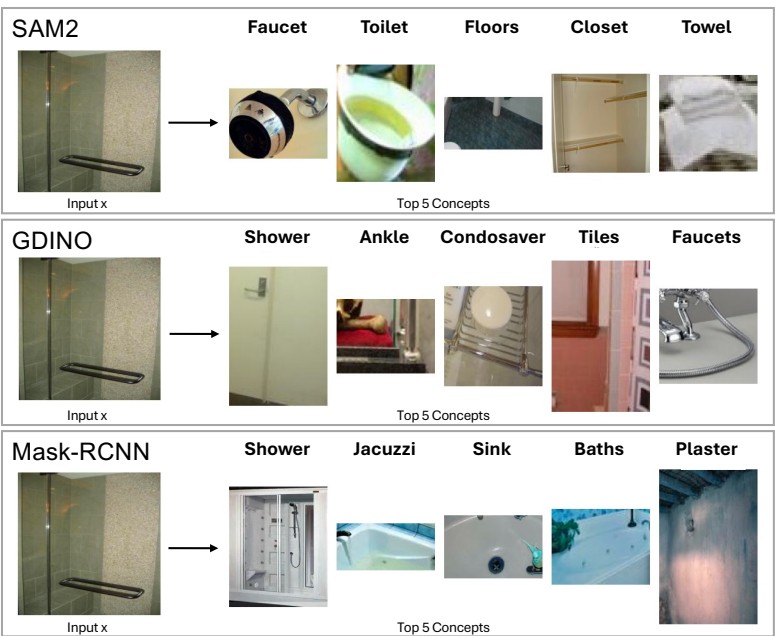

*Figure 10.* **Diverse concept proposals.** Concept sets differ in terms of granularity depending on the segmentation or detection method employed, shown on *shower* (Places365).

The choice of the concept proposal model determines the retrieved concept set. The generic SAM2 model creates a large number of concept proposals of all image elements, while Mask-RCNN returns object-centric concepts. In contrast, the

promptable GroundingDINO is specifically steered to detect common object parts. While the different concept sets only differ marginally in terms of accuracy, as shown in Table 1 and Table 2, the interpretability is highly influenced. Here, we examine on Places365 with CLIP ViT L/14 the top five image concepts in Figure 10. The figure displays interpretations for the same input image, correctly classified as *shower* by all three models, with each concept image accompanied by the closest matching textual label (g20k name space). The image consistently activates concepts that exhibit semantic alignment with the class *shower* or the larger context of the bathroom, where towels, faucets, sinks, and bath (tubs) can commonly be found. In particular, GroundingDINO (GDINO) and Mask-RCNN produce top concepts that closely match CLIP's textual embedding for the *shower*, even though both concepts appear quite different, as the Mask-RCNN concept resembles a shower, while GDINO's concept corresponds to an image of a door.

## G.2. Concept intervention

DCBM allows for the removal of specific, undesired concepts before its training. This is achieved by leveraging CLIP's multimodal capabilities: given a textual prompt, we identify and exclude visual concepts that are highly similar to the specified concept in the embedding space. For instance, in Figure 11 we remove exemplary concepts by computing the embedding of the word and discarding all visual concepts with high cosine similarity.

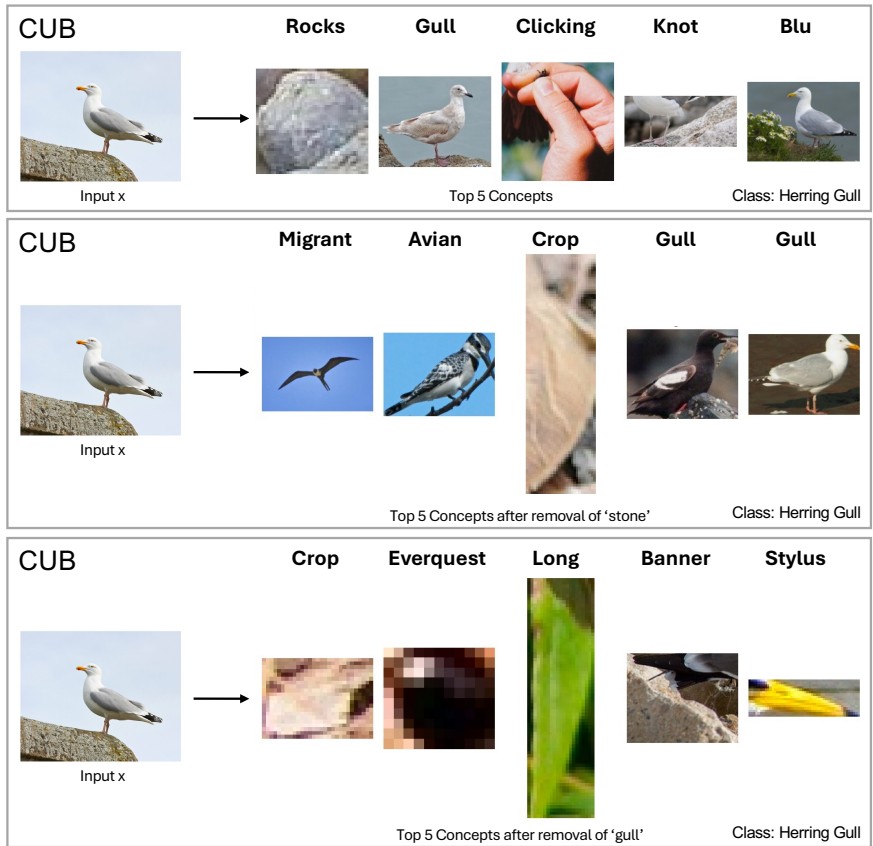

*Figure 11.* **Concept intervention in CUB.** The first instance shows the top five concepts of a gull without concept intervention. The second one shows the same instance, but with a trained DCBM with the removal of 'stone' concepts. The third one is a DCBM with the removal of the concept 'gull'.

The CBM trained with the included stone concept achieves 81.8% classification accuracy. After retraining the model without the stone-related the accuracy remains unchanged and even improves by 0.4 % with the removal of the gull-related concepts. For both examples, the explanations for the class gull no longer reference the concepts removed, demonstrating that we successfully intervened in the model concept space.

## G.3. Concept visualizations

We compare the concept-based explanations for examples from the Places365 dataset—originally used in (Rao et al., 2024)—in Figure 12 and Figure 13.

In Figure 12, two images are shown alongside explanations generated by CDM (Panousis et al., 2023), LF-CBM (Oikarinen et al., 2023), DNCBM (Rao et al., 2024), and our DCBM, which additionally includes visual explanations. For our method, we also present outputs for each of the CLIP backbones used in this study.

In the left example, labeled as "*raft*," all CBMs correctly classify the image, though the visual explanations from our DCBM vary depending on the backbone, offering different perspectives on the relevant concepts. In contrast, the right image, labeled "*swimming hole*," is misclassified by our model as "*creek*" or "*river*." However, the visual concept attributions provide insight into this decision, revealing that the confusion can be attributed to the high semantic similarity between these classes. As stated before, DCBM concepts are visual, thus more attention should be paid to the visual concept than to its textual description. We have discussed this figure in the review phase and have described there the concepts proposed by DCBM with a VIT-B/16 backbone.

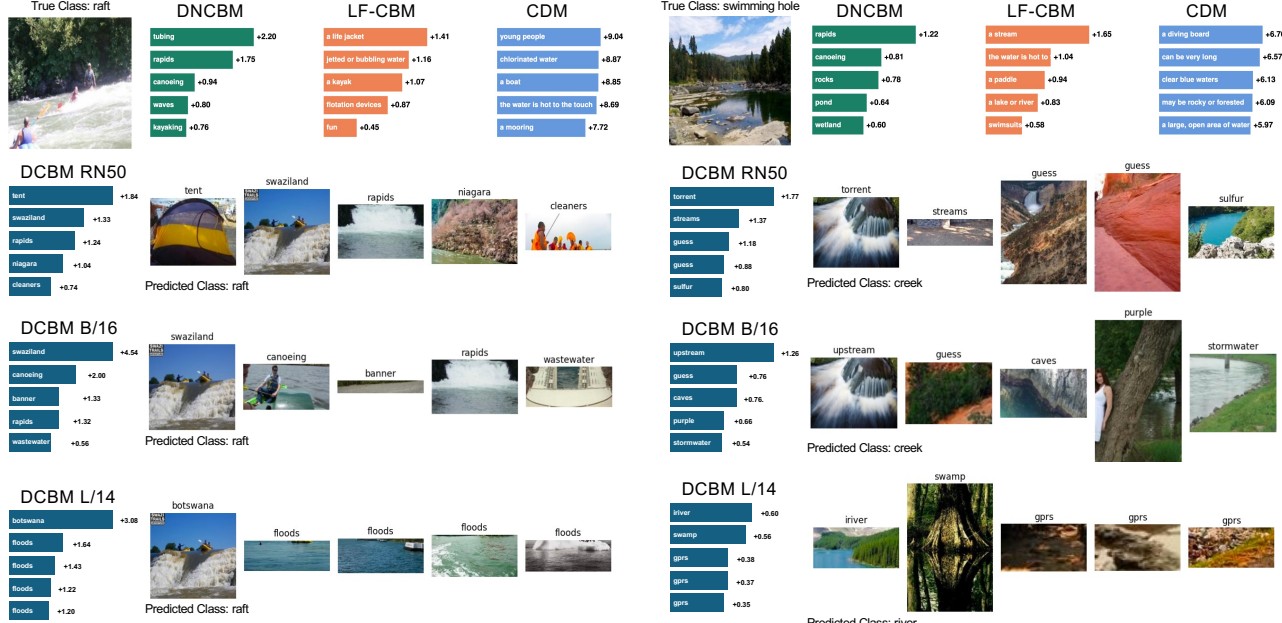

*Figure 12.* **Comparison of concept explanations across CBMs.** This figure shows concept-based explanations for two images from the Places365 dataset, as generated by four different CBMs: CDM (Panousis et al., 2023), LF-CBM (Oikarinen et al., 2023), DNCBM (Rao et al., 2024), and our proposed DCBM. The examples for CDM, LF-CBM, and DNCBM are adapted from (Rao et al., 2024).

In Figure 13, we again present examples originally shown in (Rao et al., 2024), but this time the comparison is limited to DNCBM and our DCBM. The left image, labeled as "*trench*" is correctly classified by both models. Interestingly, while DCBM assigns different weights to visual concepts depending on the CLIP backbone, the attributions remain meaningful for interpretability.

In contrast, the right image, labeled as "raceway," is misclassified by both DNCBM and most DCBM variants—except for the ResNet-50 backbone, which correctly identifies the class. In particular, ResNet-50 exhibits the lowest overall classification accuracy in the Places365 data set, but is successful in this case. The incorrect predictions, such as "auto showroom" and "auto factory", are semantically similar to "raceway", highlighting the inherent challenge of distinguishing such closely related categories. Once more, the visual concepts provide valuable insight into the semantic features each model emphasizes in its decision-making process.

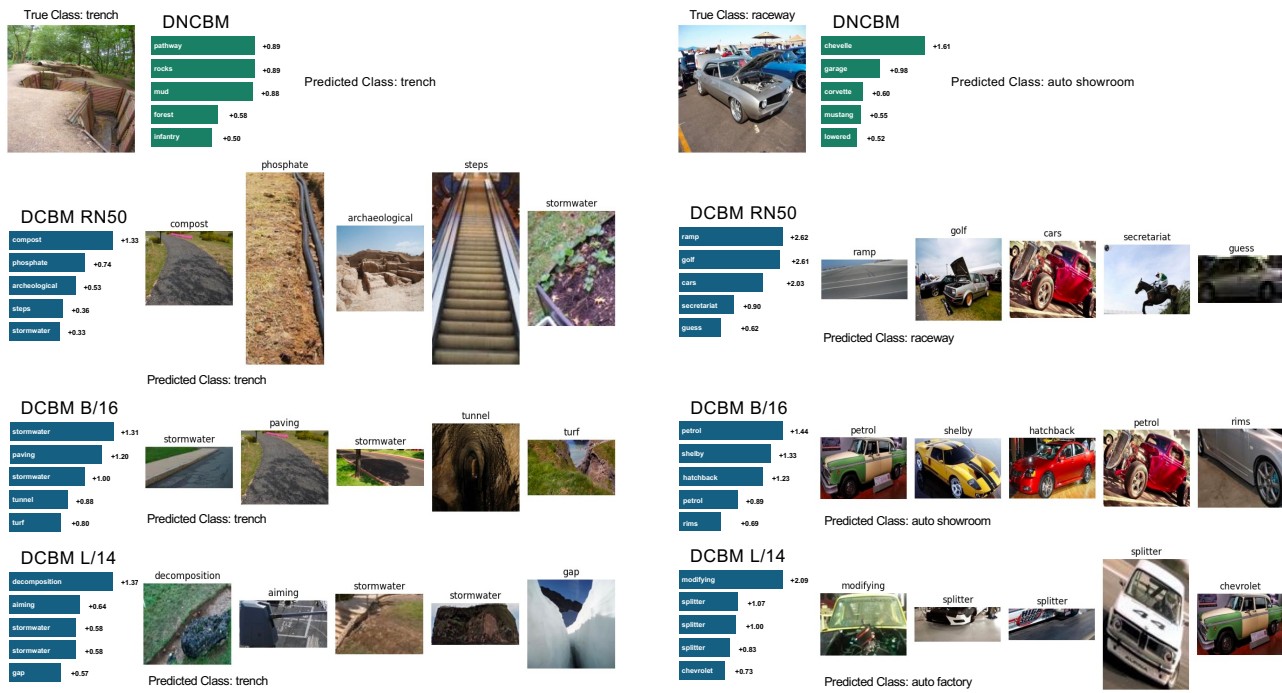

*Figure 13.* **Concept explanations from DCBM and DNCBM.** This figure presents a comparison of concept explanations for two images from the Places365 dataset, illustrating outputs from DNCBM (Rao et al., 2024) and our DCBM. For each DCBM example, we show the concept attribution across the three CLIP backbone variants used in our study.

