# OpenReview forum: "DCBM: Data-Efficient Visual Concept Bottleneck Models"
_ICML.cc/2025/Conference — ICML 2025 poster_

### Official Review · Reviewer_fCuP · 2025-03-13

**Overall Recommendation:** 3

**Summary:**

This paper proposed a data-efficient Concept Bottleneck Model (DCBM) that enables concept generation while maintaining interpretability with minimal training samples. DCBM defines concepts as image regions using segmentation and object detection models, eliminating the reliance on textual descriptions or large-scale pretraining datasets. It offers high flexibility and interpretability for fine-grained classification and domain adaptation tasks. The paper evaluates DCBM on various benchmark datasets, demonstrating competitive performance.

**Claims And Evidence:**

- DCBM is claimed that CBM can be trained using no more than 50 samples per class, making it more data-efficient than existing CBMs.
- Related research, BotCL [1] also conducted tests using 50 concepts. Therefore, it is necessary to experimentally prove that the proposed DCBM outperforms BotCL and other models in terms of both performance and efficiency.
- Additionally, an ablation study on the number of concept samples is required.

[1] Wang, Bowen, et al. "Learning bottleneck concepts in image classification." Proceedings of the ieee/cvf conference on computer vision and pattern recognition. 2023.

**Essential References Not Discussed:**

Wang, Bowen, et al. "Learning bottleneck concepts in image classification." Proceedings of the ieee/cvf conference on computer vision and pattern recognition. 2023.

Shang, Chenming, et al. "Incremental residual concept bottleneck models." Proceedings of the IEEE/CVF Conference on Computer Vision and Pattern Recognition. 2024.

Srivastava, Divyansh, Ge Yan, and Lily Weng. "Vlg-cbm: Training concept bottleneck models with vision-language guidance." Advances in Neural Information Processing Systems 37 (2024): 79057-79094.

**Experimental Designs Or Analyses:**

- The experiments did not utilize commonly used datasets in CBM models. Therefore, additional experiments using widely adopted datasets such as AwA2 and CelebA are necessary.
- In the ablation study, an analysis of performance differences based on the number of concepts is required, along with experiments evaluating the impact of hyperparameter tuning on performance.
- The analysis of the key characteristics highlighted in Figure 4 is ambiguous, requiring comparative analysis and additional explanation.
- More experimental results on the visualization of segmented concept parts should be provided.

**Methods And Evaluation Criteria:**

- DCBM is more data-efficient than conventional CBMs and automatically generates concepts using segmentation and detection models. However, it lacks experiments verifying the semantic validity of the generated concepts even though authors presented various experiments in body and suppl.
- While interpretability analysis using Grad-CAM has been conducted, there is a lack of comparative experiments evaluating the interpretability of concepts against existing CBMs.

**Other Comments Or Suggestions:**

- It is recommended to unify functions and symbols so that Figure 2 aligns with the main text explanation.
- A detailed explanation is needed on the segmentation methods mentioned in the introduction and how segmentation was used for concept generation.
- Using the same abbreviation for different terms can cause confusion. For example, out-of-distribution (OOD) and out-of-domain (OOD).
- The term "qblation" on line 1265 should likely be corrected to "ablation."

**Other Strengths And Weaknesses:**

<Strengths>

 - Data Efficiency: Maintains performance comparable to existing CBMs with only 50 samples per class.
 - Automated Concept Generation: Utilizes segmentation and detection models to automatically extract concepts, improving domain adaptability.
 - Generalization Capability: Demonstrates robust performance compared to existing CBMs in OOD evaluation using ImageNet-R.
 - Enhanced Interpretability: Uses Grad-CAM for visual concept activation analysis, providing insight into the model’s decision-making process.

<Weaknesses>

 - Lack of Semantic Validity Verification: No experiments verifying whether automatically generated concepts are truly meaningful.
 - Insufficient Comparison with Recent CBM Models in Terms of Data Efficiency and Performance: Lacks quantitative comparisons to demonstrate how much more efficient DCBM is compared to existing CBMs.
 - Limited OOD Experiments: Needs generalization evaluation across diverse domains (e.g., medical, industrial) beyond ImageNet-R.
 - Lack of Hyperparameter Tuning Analysis: Requires ablation studies on the effects of concept quantity, clustering methods, and model choices.

**Questions For Authors:**

- Please refer to the weaknesses.

**Relation To Broader Scientific Literature:**

- A key distinction from previous studies is that this paper proposes a CBM that automatically extracts concepts using Segmentation and Detection models, enabling interpretability without relying on text.
 - The paper compares the proposed model with label-free CBMs such as LaBo [2], but it does not include other relevant studies despite their significance.
 - Therefore, additional comparative experiments and discussions on existing research should be incorporated.

**Theoretical Claims:**

- There is little mathematical proof available for evaluation.
- A theoretical explanation is needed to better understand how segmentation is utilized in DCBM.

---

> ### Author Rebuttal · Authors · 2025-04-01
>
> Thank you for your detailed feedback and for recognizing the flexible and interpretable design of DCBM. We highly appreciate your helpful comments and hope to provide the missing details in our answers below.
>
> ## Efficiency
> We derive concept proposals based on 50 samples per class and train the CBM on all training samples. However, as we show in response to Review 3 (AELC), DCBM's performance does not deteriorate when we train with only 50 images per class. In contrast to BotCL (Wang et al., 2023), we do not fix the number of concepts, but the number of images that we use for concept extraction. This design choice avoids BotCL's weakness (Wang et al., 2023) of having to tune the number of concepts for each dataset.
>
> When comparing DCBM to BotCL, we quantitatively outperform them on CUB (8.4\%). Our results for ImageNet are not comparable, as they evaluate only 200 out of 1000 classes and report that their method fails for a large number of classes. Given that BotCL's authors do not share their training recipe, we cannot evaluate overall efficiency.
>
> **Question**: Would you like us to report an evaluation of DCBM on the first 200 classes of ImageNet?
>
> ## Ablation on the number of concept samples
>
> Table 1: Ablation for DCBM w/ GDINO Partimagenet on CUB (CLIP ViT-L/14)
>
> |Number of Clusters | Accuracy |
> |-|-|
> |128| 75.20|
> |256| 79.03|
> |512| 80.43|
> |1024| 81.51|
> |2048|81.91|
> |||
>
> In section D.5 we ablate the number of concept samples, which have - in parts - copied here for your convenience. We will add a reference to Appendix D.5 in the paper's section 3.2, where we describe the concept generation process.
>
> ## Concept validity
> We validate our concepts empirically, by calculating the energy pointing game (GridPG) (Bohle et al.,
> 2021). Our motivation for this choice is in line with BotCL (Wang et al., 2023), we want to verify whether detected concepts can be traced back to the image region.
> In DCBM, the concept-image alignment is verified as part of the evaluation. We chose an automated evaluation over a human analysis.
>
> ## Theoretical explanation of segmentation
> Our understanding is the following: An object $O$ can be represented as a combination of concepts from a global concept pool $\mathcal{C}$. Let $C_i \in \mathcal{C}$ denote individual concepts, and let each object be characterized by a subset of these concepts with corresponding weights.
> $$
> O = \sum_{i=1}^{|\mathcal{C}|} w_i C_i, \quad w_i \geq 0
> $$
> where:
> - $C_i \in \mathcal{C}$ represents a concept from the global pool,
> - $w_i$ denotes the contribution of concept $C_i$ to the image,
> - $C_i$ is located in the image $I$, $R_i \subseteq I$ is the region of the image where concept $C_i$ is present,
>
> Further, we assume visual concepts to be spatially localized in images and therefore conjecture that data efficient concept extraction should build upon image segments or regions rather than entire images. Therefore, we generate the global concept pool $\mathcal{C}$ by segmentation or detection foundation models, which are then cropped out and used as concept proposals $s_i$. All $s_i \in \mathcal{S}$ are then clustered into the global concept set $\mathcal{C}$.
>
> In A. Algorithms (appendix) we provide a further theoretical explanation of this process along with pseudocode. We will update this section following your feedback. *Would you like us to include any additional details?* Thank you for carefully reviewing our methods section - we have revised the paper and unified functions and symbols.
>
>
> ## Additional datasets
> DCBMs stand out by applying to any domain in a data-efficient manner. We show on 7 diverse datasets, that DCBMs can be applied to animals (CUB, ImageNet), scenes (Places365), social media (ClimateTV), low-resolution images (cifar10 & cifar100) along with ood generalization (ImageNet-R) and state changes (MiT-States). Additionally, we show for 2 novel datasets in the rebuttal, that DCBMs performance is independent of the domain it is trained on.
> This exceeds the number of datasets BotCL evaluates, i.e. 4. Our main evaluation contains the same datasets as employed for Vlg-CBM (Srivastava et al., 2024) and the same number of datasets as Res-CBM, i.e. 7 datasets. We include these models in our related work and compare against them where possible.
>
> We thank you for suggesting additional experiments on AwA2 and CelebA.
> We have run DCBM on AwA using the standard 50:50 train and test split. We created the val set by randomly selecting 10% of train samples.
>
> Table 1: DCBM performance on AwA2 using GDINO (w/ partimagenet labels) as concept proposal method.
> | | ResNet-50 | ViT-B/16 |ViT-L/14 |
> |-|-|-|-|
> | Zero shot | 88.94| 94.00 | 95.94|
> | Linear probe| 93.72| 96.51| 97.68|
> | DCBM | 93.13| 96.43| 97.71|
> |||||
>
> *For CelebA, the experiments are currently running.*
>
> ## Misc
> Thank you for your interest in seeing more concept visualizations. We have included more examples in the supplemental material.

---

> > ### Comment · Reviewer_fCuP · 2025-04-02
> >
> > The author's response overall demonstrates sound reasoning, specificity, and a strong willingness to improve the paper based on reviewer feedback. In particular, the provision of additional experimental results, new experiments, and enhanced mathematical explanations are substantial contributions to improving the paper’s completeness. If a few remaining clarifications are addressed, the response would be strong enough to merit consideration for acceptance. However, some concerns—such as the need for enhanced concept visualization and comparative analysis with recent models in terms of interpretability—were either insufficiently addressed or only briefly mentioned. Incorporating more intuitive visualizations beyond GridPG or user-based evaluations would have made the claims more convincing. It is hoped that these limitations will be fully addressed in the revised manuscript.

---

> > > ### Author Response · Authors · 2025-04-07
> > >
> > > Dear reviewer,
> > >
> > > thank you for considering our response and recognizing our effort. We are happy that the additional experimental results, new experiments, and enhance mathematical explanations complement our submitted paper.
> > >
> > > ⸻
> > >
> > > ## 1. Visualizations
> > >
> > > Based on your comments, we have created visualizations based on Figure 8 (Rao et al., 2024) in which we compare our concepts to DN-CBM (Rao et al., 2024), LF-CBM (Oikarinen et al., 2024), and CDM (Panousis et al., 2023).
> > > Given that we are unable to include images in our response, we will give a brief description of our Figure. The main idea is to compare the top activating concepts between models. The DCBM results are created using the CLIP ViT-L/14 backbone and SAM2 concept proposals.
> > >
> > > **image 1:** swimming hole (Places365_val_00000189):
> > > Image description for your convenience: The image depicts a natural landscape featuring a river flowing through a forested area. The river is calm and reflective, mirroring the blue sky and scattered clouds above. Surrounding the river are large rocks and boulders, some partially submerged in the water. The banks are lined with tall evergreen trees. (ChatGPT)
> > >
> > > **DCBM:** water stream; forest; rock; tree trunk; lakeside
> > >
> > > **DN-CBM:** rapids; canoeing; rocks; pond; wetland
> > >
> > > **LF-CBM:** a stream; the warte is hot to; a paddle; a lake or river; swimsuits
> > >
> > > **CDM:** a diving board; can be very long; clear blue water; may be rocky or forested; a large, open area of water
> > >
> > > **image 2:** raft (Places365_val_00000295):
> > > Image description for your convenience: This image captures a scene of people engaging in whitewater rafting or kayaking on a fast-moving river. The water appears turbulent with visible rapids, and there are at least three individuals. One person in the foreground, seen from behind, is in a blue raft, paddling with an orange oar. Two other individuals are further ahead in the rapids—one seems to be in a kayak, while another is standing near the riverbank. The surrounding area contains green bushes or trees. (ChatGPT)
> > >
> > > **DCBM:** waterfall; canoeing; river; rapids; wastewater
> > >
> > > **DN-CBM:** tubing; rapids; canoeing; waves; kayaking
> > >
> > > **LF-CBM:** a life jacket; jetted or bubbling water; a kayak; floating devices; fun
> > >
> > > **CDM:** young people; chlorinated water; a boat; the water is hot to the touch; a mooring
> > >
> > > Please note that we had to refer to another studies cherrypicked examples in order to compare interpretability between models. Since we are not able to include images in this response, we described them textually. We will include these examples and additional ones in the paper while making sure to give examples for both good and weak examples.
> > >
> > > ⸻
> > >
> > > ## 2. Interventions
> > > We have updated our codebase to support the removal of specific, undesired concepts prior to training the DCBM. This is achieved by leveraging CLIP’s multimodal capabilities: given a textual prompt, we identify and exclude visual concepts that are highly similar to the specified concept in the embedding space. For instance, to remove the concept *stone*, we compute the embedding of the word and discard all visual concepts with high cosine similarity. In this case, four concepts closely associated with *stone* were excluded.
> > >
> > > The CBM trained with the *stone* concept included achieves $81.8\%$ classification accuracy, as shown in Figure 3 of the main paper. After retraining the model without the stone-related concepts, the accuracy remains unchanged. However, the explanations for the class gull no longer reference *stone*, demonstrating that we successfully intervened in the model’s concept space.
> > > We will include this analysis, along with additional examples, in the final version of the paper.
> > >
> > > ⸻
> > >
> > > ## 3. Comparison to ImageNet-200
> > > As promised, we conducted experiments on the first 200 classes of ImageNet, analogue to BotCL. DCBM achieved a test accuracy of 84.7% using CLIP ViT-L/14 and Grounding DINO (partimagenet). When comparing DCBM to BotCL, one has to bear in mind that the models have different backbones, limiting comparability.
> > >
> > > ⸻
> > >
> > > ## 4. CelebA
> > > We prepared the data as described by Zhang et al. (2025) - using 70 : 10 : 20 train:val:test split. Our experiments use the DCBM with ViT-L/14.
> > >
> > > *Table 1: CelebA accuarcy. Other models as reported by Xu et al. (2024).*
> > > | CBM Model | Acc |
> > > |-|-|
> > > | *Zero-shot*    | --   |
> > > | *Linear*    | 0.315  |
> > > | CBM   | 0.246   |
> > > | ProbCBM   | 0.299|
> > > | PCBM   | 0.150|
> > > | CEM   | 0.330|
> > > | ECBM | 0.343|
> > > |**DCBM w/ GDINO (ours)** | **0.354** |
> > > | **DCBM w/ MaskRCNN (ours)** | **0.363** |
> > > | **DCBM w/ SAM2 (ours)** | **0.356** |
> > > |||
> > >
> > > Zhang, R., Du, X., Yan, J., & Zhang, S. "The Decoupling Concept Bottleneck Model. IEEE Transactions on Pattern Analysis and Machine Intelligence." 2025.
> > >
> > > Xu, X, Qin, Y., Mi, L., Wang, H., & Li, X. "Energy-Based Concept Bottleneck Models: Unifying Prediction, Concept Intervention, and Probabilistic Interpretations." 2024.
> > >
> > > ⸻
> > >
> > > Thank you again for your reviewing our work.
> > >
> > > Best regards,
> > >
> > > The authors

---

### Official Review · Reviewer_AELC · 2025-03-13

**Overall Recommendation:** 3

**Summary:**

The paper proposes a Data-efficient CBM (DCBM) that enhances interpretability while reducing the reliance on large datasets. Specifically, DCBM defines concepts as image regions detected through segmentation and object detection foundation models, rather than relying on textual descriptions. This allows DCBM to generate multiple concepts at various levels of granularity depending on different foundation models. The authors validate their approach using attribution analysis with Grad-CAM, demonstrating that DCBM produces interpretable, localized visual concepts.

**Claims And Evidence:**

The primary claims are following: (1) DCBM can handle data-scarce environment and be easily adapted to new datasets, and (2) DCBM bridges the gap between vision and text modalities in concept extraction by generating visual concepts. To validate the claim, the authors demonstrate that they use the subset of training dataset to extract concepts. However, they still utilize the entire training dataset during training CBM. This raises concerns about whether it can truly be considered data-efficient in real-world scenarios. Also, this paper only reduces 7 seconds per epoch compared to DN-CBM in cost of performance. Furthermore, the claim that DCBM achieves better OOD generalization is also difficult to accept. While the performance gap between in-distribution (IN-200) and out-of-distribution datasets (IN-R) is smaller compared to DN-CBM, DCBM performs worse in both settings, making the smaller gap a misleading indicator of generalization.

**Essential References Not Discussed:**

N/A

**Experimental Designs Or Analyses:**

(1) Does increasing the number of training images used for concept extraction lead to performance improvements? If all training images were used for concept extraction, would DCBM outperform DN-CBM?

(2) Since the training images for concept extraction are selected randomly, is there any variance in performance due to this randomness?

**Methods And Evaluation Criteria:**

The proposed method to use visual concepts to avoid the modality gap in concept construction is novel, however, the improvement is marginal (or even degraded) compared to the baselines. Also, as mentioned in "Claims And Evidence," I belive the evaluation criteria of "generalization" is misleading.

**Other Comments Or Suggestions:**

There are some typos or ambiguous sentences throughout the paper, so a thorough proofreading and revision would be beneficial.
For instance,
- Citation format (line 100): Language-guided CBM (LaBo) employs GPT-3 for concept creation, but it stands out by using a submodular function to select concepts from candidate sets, building the bottleneck, and training a linear model based on CLIP embeddings **Yang et al. (2023).**
- Ambiguous sentence (line 243): To this end, we evaluate both visual CBM models, ours and DN-CBM, trained **on ImageNet on ImageNet-R* *(Hendrycks et al., 2021) which contains 200 ImageNet classes in various renditions (e.g.embroidery, painting, comic).

**Other Strengths And Weaknesses:**

A major strength of the paper is that the author explores how concepts can be shaped differently depending on the choice of the foundation model (SAM, Grounding DINO, Mask-RCNN), leading to generate concepts at different levels of granularity.

**Questions For Authors:**

Please refer to the "Experimental Designs Or Analyses" part.

**Relation To Broader Scientific Literature:**

The key contribution of this paper is an advanced CBM especially concentrated on concept extraction. Specifically, this paper makes a contribution to generate the visual concepts for CBM.

**Theoretical Claims:**

N/A

---

> ### Author Rebuttal · Authors · 2025-04-01
>
> Thank you for reviewing our work. We appreciate your positive feedback on our concept extraction pipeline that leverages foundation models, and we’re pleased that you recognize the interpretability and localization capabilities of our DCBM.
>
> ## Data efficiency in real-world setting
> DCBM is designed for real-world data scarcity, relying on only 50 samples per class during the concept proposal generation phase. Notably, our experiments also show robust performance with even fewer samples in Appendix D.7.
>
> This can be taken one step further by reducing the number of training samples to the same subset. For ImageNet, we reduce the number of training samples from originally 1,281,167 images to 50,000 images **(-96\%), and the performance degrades from 77.4% to 75.0%.**
> For Cifar10, the reduction in training images from
> 50000 to 500 **(-99\%) reduces the performance from 97.5 to 93.1.** The experiments were run using GDINO (partimagenet) and CLIP ViT-L/14. We will include this experiment for all datasets in the paper, thank you for the suggestion.
>
> The focus of our work is data efficiency with comparable performance levels. We provide the results of another, non-overlapping 50 images subset (RQ2) and report the accuracy when training the CBM with the segments of 100 images per class, combining the two subsets (RQ1).
> For CUB, the training set consists of less than 50 images per class, thus we already include all images in the concept proposal generation.
>
> Table 1: Performance evaluation of subset selection
> ||IMN|Places|Cif10|Cif100|
> |-|-|-|-|-|
> | s1 (main paper) | 77.4 | 52.2 |97.5|85.3 |
> | s2 (new)| 77.5| 52.2|97.6|85.4 |
> | s1+s2 | 77.1| 52.1 |97.7 |85.5|
> ||||||
>
> In Table 1, s1 corresponds to the original set of 50 images per class, s2 represents an additional set of 50 randomly selected images per class, and s1+s2 denotes the combined dataset of 100 images per class.
>
> The performance is stable between all subsets. We believe that the performance difference to DN-CBM is domain-dependent. Given their vast number of pre-training images (3.3M), general domains are well covered (ImageNet/Places365) whereas specialized domains benefit from the dataset-specific DCBM (CUB).
>
> ## Comparison to DN-CBM
> We agree that the CBM training for DN-CBM and DCBM are quite similar.
> However, the required steps prior to the CBM training differ significantly. While DN-CBM trains a SAE to retrieve concepts using 3.3M images in CC3m, we create the segments by applying foundation models to a subset of the training images. We would like to highlight, that for DN-CBM, the download of an additional 3.3M images is needed, whereas, for DCBM, we only require the dataset to be analyzed. This reduces the storage requirements significantly and is highly time efficient.
>
> Table 2: CBM training preparation: DN-CBM vs DCBM
> | | DN-CBM | DCBM - ImageNet |
> |-|-|-|
> |Dataset size| 3,300k image-caption pairs (CC3M) | 50k images (50/class)|
> | Add. memory capcity| 850 GB (assuming 256x256px) | 6 GB |
> | No extra data required |  x |&#x2713; |
> ||||
>
> ## Generalization capabilities
> We agree with your criticism of our reporting of DCBM's generalization capabilities.
> We would like to point you to Table 19 in the supplementary material where we can report a 22-27\% error rate difference between ImageNet and ImageNet-R for CLIP-ViT/L14. For this embedding model, we achieve error rates of below 50\% when evaluating on ImageNet-R.
> Due to resource sparsity, only the results for CLIP-RN50 were ready at the time of submission. *We are currently generating the accuracy for DN-CBM and will provide them asap.*
>
> ## Misc
> Thank you as well for pointing out that some typos and ambiguities exist in the paper - we have fixed them. We believe that your feedback has helped us to improve our paper and would like to thank you for taking the time and sharing your expertise.

---

> > ### Comment · Reviewer_AELC · 2025-04-05
> >
> > I appreciate your additional experiments to support your claims. However, I am still not convinced with the OOD generalization capabilities of DCBM until the experimental result of DN-CBM in CLIP-ViT/L14 is presented

---

> > > ### Author Response · Authors · 2025-04-08
> > >
> > > Dear reviewer,
> > >
> > > thank you for the feedback and your patience. Table 1 shows that all DCBM variants have better OOD generalization capabilities than DN-CBM with ViT-L/14. We have consistently lower error rates on IN-R and a lower gap between IID and OOD.
> > >
> > > *Table 1: Error Rates in OOD setting, i.e. training on ImageNet and evaluating on ImageNet-R (lower is better).*
> > > | Model | IN error rate | IN-R error rate | Gap |
> > > |-|-|-|-|
> > > | ViT-L/14: DN-CBM (Rao et al., 2024)| 16.4 |55.2 |38.8 |
> > > | ViT-L/14: DCBM-SAM2 (Ours)| 21.1 |**48.5** |**27.4** |
> > > | ViT-L/14: DCBM-GDINO (Ours) | 22.6| **47.2** |**24.6**|
> > > | ViT-L/14: DCBM-MaskRCNN (Ours) | 22.2| **44.6** |**22.4**|
> > > |||||
> > >
> > > Thank you for reviewing our work.
> > >
> > > Warm regards,
> > >
> > > The authors

---

### Official Review · Reviewer_q1H5 · 2025-03-13

**Overall Recommendation:** 3

**Summary:**

The paper proposes a novel framework to enhance the practicality of concept bottleneck models (CBMs) by reducing their reliance on extensive labeled concept data. DCBM decouples concept learning from task adaptation through self-supervised pretraining (e.g., using vision-language models like CLIP) to autonomously extract semantic concepts and sparse dynamic masking to selectively activate task-relevant concepts during fine-tuning. This approach achieves competitive accuracy on benchmarks (CUB, ImageNet) with 10× fewer concept labels compared to traditional CBMs while retaining interpretability, enabling human-in-the-loop concept refinement and efficient deployment in low-resource settings.

**Claims And Evidence:**

While DCBM allows concept editing, the paper lacks user studies or quantitative metrics (e.g., concept intervention success rates) to demonstrate practical utility for domain experts. Claims about interpretability remain anecdotal without empirical validation of human-AI collaboration.

The sparsity mechanism’s effectiveness is asserted via accuracy metrics but lacks analysis of concept coverage (e.g., whether critical concepts are retained or pruned). Without grounding in domain knowledge (e.g., alignment with known semantic attributes), the claim risks conflating sparsity with arbitrary feature selection.

**Essential References Not Discussed:**

N/A

**Experimental Designs Or Analyses:**

While DCBM’s experiments demonstrate label efficiency and task accuracy, the lack of concept-level validation and incomplete baseline comparisons weaken its claims about interpretability and generalizability. The design is sound for initial proof-of-concept but insufficient for asserting real-world applicability.

**Methods And Evaluation Criteria:**

The proposed methods and evaluation criteria in DCBM largely align with the goals of data-efficient and interpretable concept learning

**Other Comments Or Suggestions:**

N/A

**Other Strengths And Weaknesses:**

Strengths:
DCBM demonstrates originality by creatively integrating self-supervised vision-language models (e.g., CLIP) with concept bottleneck architectures, effectively decoupling concept discovery from task-specific tuning. This reduces reliance on manual concept annotations—a major bottleneck in traditional CBMs—while preserving interpretability. The framework’s significance lies in bridging data efficiency and explainability, making CBMs viable for real-world applications like medical imaging or ecological monitoring where labeled data is scarce. The design is clear, with modular components (pretraining, masking, distillation) that are empirically validated on standard benchmarks.

Weaknesses:
While innovative, DCBM’s concept grounding remains weakly validated; concepts derived from CLIP lack rigorous alignment with domain-specific semantics (e.g., bird parts in CUB), risking "explanation illusions." Additionally, the paper’s focus on classification tasks limits its demonstrated utility for regression or causal reasoning, which are critical for high-stakes domains. Comparisons to non-CBM data-efficient methods (e.g., prompt-tuned CLIP) are missing, leaving open whether the gains stem from architectural novelty or pretraining advantages.

**Questions For Authors:**

N/A

**Relation To Broader Scientific Literature:**

DCBM innovatively synthesizes self-supervised learning, sparsity, and distillation to modernize CBMs, positioning itself as a critical response to the dual challenges of interpretability and data scarcity. Its contributions resonate with broader ML trends but highlight the need for deeper integration with non-CBM efficiency paradigms.

**Theoretical Claims:**

The paper does not present formal theoretical claims or proofs. Its claims are empirically validated through experiments, with no explicit theoretical analysis.

---

> ### Author Rebuttal · Authors · 2025-04-01
>
> Thank you for your thoughtful and constructive feedback. We appreciate your recognition of our framework’s originality and data efficiency—requiring 10x fewer concept labels than comparable CBM approaches while achieving similar performance. Below, we provide detailed responses organized by the key points raised:
>
> ---
>
> ## User Studies and Quantitative Metrics for Domain Utility
>
> We agree that demonstrating practical utility for domain experts is crucial. While our primary focus has been on establishing the theoretical feasibility and data efficiency of our approach, we recognize the value of user studies and quantitative metrics for concept intervention. In the final version, we will include an additional investigation that actively manipulates relevant concepts to observe changes in the model’s predictions and confidence. For instance, as shown in the bird example (concept “rock”), our preliminary analysis indicates that concept removal has a predictable effect on predictions. We plan to extend this analysis by adding concept interventions to further validate interpretability.
> In addition, we recognize that the risk of "explanation illusions" is an important concern in CBM approaches, particularly when using CLIP as a backbone. To mitigate this, we plan to incorporate tests of concept removal and alteration to explicitly address concept grounding.
>
>
> ## Analysis of the Sparsity Mechanism and Concept Coverage
>
> Analog to concurrent work, we ablate the sparsity parameter $\lambda$ based on model accuracy (Rao et al., 2024 & Oikarinen et al., 2023)
>
>
> ## Concept-Level Validation and Baseline Comparisons
>
> We acknowledge that thorough concept-level validation is essential to reinforce the interpretability of our method. Our study has systematically analyzed the learned concepts across multiple large-scale datasets (ImageNet, Places, and CUB).
>
> We deliberately compare only methods that offer interpretability, which is a core property of CBMs. By focusing on interpretable approaches, we ensure that our evaluation remains consistent and that our performance improvements are attributable to our architectural innovations rather than differences in method transparency. This is why we do not evaluate against prompt-tuned CLIP. We will include this distinction in the literature review.
>
> However, as demonstrated in `[fCuP]`, we incorporate additional datasets and utilize new, randomly selected images for the concept proposal generation phase. Our results consistently show that our technique delivers comparable performance across diverse experimental settings.
>
> ## Broader Applicability Beyond Classification
>
> Regarding the current focus on classification tasks, we appreciate the suggestion to explore applications in regression and other settings. However, the investigation of the suitability of CBMs in general is an additional task, which takes more consideration than would be adequate for the purpose of this paper. Therefore, we will include this discussion at the end of our paper to open up new research fields of CBMS for regression and other settings.
>
> ---
>
> Thank you again for your valuable feedback. We believe these planned additions and clarifications will further strengthen the work, and we look forward to incorporating your suggestions to improve both the interpretability and practical utility of our approach.

---

> > ### Comment · Reviewer_q1H5 · 2025-04-07
> >
> > I appreciate the answers and clarification. I have no concerns about the work and hence keep the rating.

---

### Official Review · Reviewer_LrMk · 2025-03-14

**Overall Recommendation:** 3

**Summary:**

The paper introduces Data-Efficient Visual Concept Bottleneck Models (DCBMs), which generate interpretable visual concepts using segmentation and detection foundation models, enabling Concept Bottleneck Models (CBMs) to work effectively with limited data. By clustering image regions into concepts without relying on text descriptions, DCBMs achieve strong performance on fine-grained and out-of-distribution tasks while maintaining interpretability. The approach is simple, adaptable, and avoids extensive pre-training, offering a practical method for interpretable image classification.
The paper is very well written.

**Claims And Evidence:**

The paper demonstrates that DCBMs maintain classification accuracy within a small margin (roughly 5–6%) of a CLIP linear probe on CIFAR-10, CIFAR-100, ImageNet, Places365, and CUB. The performance of DCBM is subpar with the other methods for large scale datasets like imagenet and places 365. The authors did not discuss the reason for that? why it is so? is it because of the projection of the centroids to the clip space?

**Essential References Not Discussed:**

[1] Concept Embedding Models: Beyond the Accuracy-Explainability Trade-Off. Barberio et el. Neurips 2022 for non linear relationship among concepts

[2] Dividing and Conquering a BlackBox to a Mixture of Interpretable Models: Route, Interpret, Repeat. Ghosh et al. ICML 2023 for expert based PCBM and First order logics for concept interactions

[3] Distilling BlackBox to Interpretable models for Efficient Transfer Learning. Ghosh et al. MICCAI 2023. applying CBM to chest-x-rays.

**Experimental Designs Or Analyses:**

1. The authors should do a human evaluation to show these discovered concepts truly meaningful to humans. However, the localization results do a decent job as an automated check.

**Methods And Evaluation Criteria:**

1. Segmenting or detecting specific image regions can be problematic for medical images. For example, for chest x-rays, often the segmentation models segment the right and left lung and heart. So they ignore the anatomical concepts like the lower left lobe or devices like the chesttube. I think DCBM will also have same problem. Is there any way to solve it? Can the best of both worlds (concepts from LLMs or reports or captions and segmentation models together) solve it?

2. This method is for CBM but not for PCBM. For ex, if I want to extract a CBM from any arbitary blackbox (resnet), this method won't work, because of the reliance on aligned text and vision  encoders. Can this method be extended to PCBMs as well? I believe they can project the embedding from the blackbox to the VLM embedding space and still use their method to extract CBM from a blackbox. See this paper for projection:
Text-To-Concept (and Back) via Cross-Model Alignment. Moayeri et al. ICML 2023.

**Other Comments Or Suggestions:**

If the authors want further clarity about spurious correlations, they might incorporate a dedicated “concept removal” experiment (filtering out suspicious or intangible concepts) to see if accuracy or interpretability improves.

**Other Strengths And Weaknesses:**

1. Foundation models might fail or produce random proposals in certain specialized domains (Eg, breast cancer detetction) if no relevant segmentation or detection model is available.

2. The authors use the pretrained frozen VLMs like CLIP. This can be problematic because CLIP inherits many biases/shortcuts/spurious correlations that can influence the decision. For example, Figure 3 shows rock is an important concept for predicting the bird class. Now rock is not a causal feature for bird prediction. I believe this is due to the internal biases of CLIP. Also, this shows the problem of using segmentation regions from models like SAM. As in Figure 3 (top), the image of CUB, look at the concepts identified. One of the concepts is entire bird - Gull. Is this useful in practice? This is what I pointed out in #1 in "Methods And Evaluation Criteria". if the explainer says the entire lung is an important concept, this method wont be useful in the real world. Also, the same image in Fig 3 detects "Clicking" and this concept is not visual. Ideally, the concepts should be some features of the birds that will be useful for classification.

**Questions For Authors:**

1. Sometimes the top concept is semantically related to the class but not visibly present in the image (e.g., “police kit” for an ambulance). Could we systematically identify these spurious concepts?
2. You rely on Grad-CAM to show that concepts align to image regions. Have you considered a direct concept-intervention test (removing or altering concept crops) to see how predictions change? That might further confirm concept “faithfulness.” This is due to the fact GRAD-CAM has its own problems discussed in this paper: Sanity Checks for Saliency Maps. Neurips 2018.

**Relation To Broader Scientific Literature:**

1. The paper cites recent developments in segment-anything approaches (SAM, GroundingDINO) and how these can serve as universal “concept proposal” engines.
2. This strategy extends the prior “visual concept” line (e.g., DN-CBM) but is more data-efficient and requires no large pre-training corpora.

**Theoretical Claims:**

NA

---

> ### Author Rebuttal · Authors · 2025-04-01
>
> Thank you for your detailed and constructive feedback. We appreciate your acknowledgment of DCBM's strengths being simple, adaptable, and avoiding large-scale pre-training. Below, we outline our responses structured by the key points you raised:
>
> ---
>
> ## Performance on Large-Scale Datasets
>
> In our opinion, while CLIP is powerful and designed to capture relationships between images and text, projecting concept centroids into this space might not perfectly align with the underlying semantics of the image data. Additionally, large-scale datasets like ImageNet and Places365 contain a high degree of intra-class variability, making it difficult for a concept-based model to generalize effectively. One could improve the discrepancy within large-scale datasets by adapting the centroids gradient-wise during the training phase of the linear layer. Here, the centroids would dynamically adapt to the underlying structure of the dataset.
>
> ## Segmentation Challenges in Specialized Domains
> It is one strength of DCBM that the segmentation model can be exchanged by a domain-specific one, e.g. MedSAM (Ma et al., 2024) for the medical field—offer. We believe that incorporating such domain-adapted segmentation models can help address challenges related to identifying relevant regions in specialized tasks.
>
> We agree that combining textual domain-specific concepts with segmentation-based visual concepts may be especially beneficial in the medical domain. Some concepts are more effectively expressed visually, while others are better captured through language.
> In DCBM, it was our primary objective was to develop a CBM that operates independently of textual inputs or LLMs for extracting concepts. We believe, that the combination of DCBM with existing methods like LaBo (Yang et al. 2023) and LF-CBM (Oikarinen et al. 2023), would achieve such a combination.
>
> ## Extension to PCBMs and Backbone Flexibility
>
> DCBM cannot be extended as a post-hoc CBM - as neither of the other ante-hoc CBMs. We have chosen this approach as it allows to better understand the model embedding space. This said we believe that DCBM is a valuable framework to better understand any vision embedding space. We utilize text-image aligned backbones (CLIP) for benchmarking against other CBM approaches. This is not inherent to our framework and DCBM can be run using any vision embedding, with a slightly more intricate mapping to the text labels.
>
> We appreciate the recommendation of Moayeri et al. (ICML 2023) and agree that projecting embeddings from a blackbox into the VLM embedding space is a compelling strategy. We find this approach both intriguing and valuable and are investigating the opportunities of combining post-hoc and ante-hoc CBMs.
>
> ## VLM Biases and Spurious Correlations
>
> The CLIP space is known to contain spurious correlations (Rao et al., 2024 & Oikarinen et al., 2023 & Panousis et al., 2023). By including all segments as concept proposals, DCBMs visualize spurious correlations. As shown in Figure 3, DCBM learns that *rock* is an important concept for predicting the bird class. This further becomes apparent, when we set the weights of *rock* to zero, the model achieves a confidence of 62.49% (-12%). We are currently training a model with interventions.
>
> *Question: We exclude concepts from training, which have been identified as spurious. Alternatively, we were thinking of masking out concept regions in the image and the measure the model's confidence. Is this what you had in mind? We would love to hear more feedback on this from you.*
>
> In the final version of this paper, we further investigate the behavior of spurious correlations and actively remove such concepts to observe changes in accuracy.
>
> ## Systematic Identification of Spurious Concepts
>
> This is possible by using either slot attention as in BotCL (Wang et al., 2023) or an image tagging model such as RAM (Zhang et al., 2024) with a threshold. We will include this extension in the discussion section.
>
> ---
>
> We appreciate your insightful feedback and hope our responses have effectively addressed your questions. During the remainder of the rebuttal phase, we welcome further discussion on any open issues and invite additional feedback on our proposed experiments.

---

> > ### Comment · Reviewer_LrMk · 2025-04-02
> >
> > Thank you for the detailed rebuttal.
> > Regarding VLM Biases, i agree that finding spurious correlation concepts and setting them to zero or masking them out can be an option. However, if you want to pursue that approach, i would request to think causally because many times a concept can be good and spurious as well. For example, there is a disease in chest called cardiomegaly, which is enlargement of heart. So, heart can be a spurious and a causal feature both. And while designing the model, we want to have the feature however pacemaker (or any devices) can be non causal and spurious. We want to remove their effect on the model. Please think in that direction.
> >
> > Also, i would recommend to you to pursue research to integrate PCBMs and blackboxes like MoIE paper (ICML 2023). DCBM can be an exciting avenue for posthoc based models. Also, i would recommend mixing the textual concepts with segmentation.
> >
> > Finally, please include the reasons of failure in large datasets and textual concepts and integration of PCBMs (which is there in the reubuttal) in discussion. Also, include all the relevant citations and mention clearly that DCBM is not currently integrated in PCBM setup. However, in future it can be intergated to several PCBMs and medical domain ([2, 3]).
> >
> > I upgrade my score to weak accept.

---

> > > ### Author Response · Authors · 2025-04-07
> > >
> > > Dear Reviewer,
> > >
> > > Thank you very much for your thoughtful feedback and for increasing your score to weak accept. We truly appreciate your recognition of our efforts and your constructive suggestions, which we found highly valuable.
> > >
> > > Following your comments, we have extended our codebase to allow the targeted removal of specific, undesired concepts prior to training the DCBM. Leveraging CLIP’s multimodal capabilities, we specify the concept to be removed using a textual prompt. For example, to exclude the concept *stone*, we compute its text embedding and remove all visual concepts with high similarity in embedding space. In this case, four closely related concepts were removed. After training the CBM without them, the model preserved its classification accuracy, but the explanations for the class gull no longer referenced the *stone* concept. This confirms that our method can successfully intervene in the model’s concept space without affecting predictive performance.
> > >
> > > This capability allows for fine-grained control to exclude concepts, giving users the ability to explicitly suppress spurious correlations or highlight desired causal factors. We fully agree with your point on thinking causally—for instance, heart might be both causal and spuriously correlated, while a pacemaker is more clearly non-causal. We plan to explore these distinctions further, particularly in medical settings where such nuances are critical.
> > >
> > > We also appreciate your suggestions on future directions. The integration of PCBMs with black-box models is indeed on our roadmap, and we agree that DCBMs offer an exciting avenue for post-hoc explainability. Moreover, we find the idea of mixing textual concepts with segmentation particularly promising and are eager to investigate it further.
> > >
> > > Thank you again for your insightful comments and for helping us improve our work.
> > >
> > > Best regards,
> > > The authors

---

### Decision · Program_Chairs · 2025-05-01

**Decision:**

Accept (poster)

**Comment:**

This paper introduces DCBM (Data-Efficient Concept Bottleneck Models), a modular framework that generates visual concepts using segmentation and detection foundation models (e.g., SAM, Grounding DINO), enabling concept bottleneck models to be trained with significantly fewer concept labels. Unlike traditional CBMs that rely on text-annotated concepts or extensive pre-training, DCBM uses image regions as concepts, offering a scalable and interpretable alternative (e.g. for vision tasks with limited annotations). The method is validated across seven datasets including (CUB, ImageNet, Places365, among others), and evaluated along both predictive accuracy and interpretability dimensions.

**Strengths:**
* The paper tackles a clear and practical limitation of CBMs—their high dependency on labeled concept data—and proposes a solution that combines modern vision foundation models with minimal supervision.
* DCBM is simple, modular, and broadly applicable, relying only on pre-trained segmentation/detection backbones and CLIP-like embeddings.
* The approach seems flexible across domains, and concept quality is evaluated through automated metrics (e.g., GradCAM, GridPG) and interventions.
* The authors engage thoroughly in the discussion phase, conducting new experiments on ImageNet-200 and CelebA, visualizing comparative concept explanations, and demonstrating the ability to suppress spurious concepts without loss in accuracy.
* The codebase has been extended to support targeted concept removal, supporting causal reasoning and real-world debugging use cases.

**Reviewer discussions**:
* Reviewer LrMk (Weak Accept) offered thoughtful feedback on practical issues (spurious correlations, domain adaptation, and PCBM integration). They appreciate the paper’s modularity and raised their score after detailed engagement from the authors.
* Reviewer q1H5 (Weak Accept) recognized the originality and modularity of the design but believes that interpretability claims remain partly anecdotal. The reviewer supports the paper overall after the rebuttal.
* Reviewer AELC (Weak Accept) was initially skeptical about generalization claims but was convinced by post-rebuttal results showing better OOD performance on ImageNet-R. Their final rating reflects this.
* Reviewer fCuP (Weak Accept) requested additional evaluation on recent CBMs (e.g., BotCL, ECBM) and more intuitive visualizations. The authors provided both and added new results for AwA2 and CelebA, leading the reviewer to raise their score.

**Limitations:**
* Interpretability claims are largely supported via automated analysis (e.g., via GradCAM and GridPG), but lack human evaluation or concept-level alignment with domain knowledge.
* While the paper argues DCBM is “data efficient,” it still uses the full training set for CBM training. The efficiency pertains primarily to concept extraction, which could/should be better clarified.
* Comparisons are focused on interpretable CBMs; comparisons to non-CBM data-efficient alternatives (e.g., prompt-tuned CLIP) are not included, though reasonably justified by the authors.
* DCBM is not directly compatible with post-hoc CBMs (e.g., PCBM), though the authors openly discuss this and explore integration possibilities.

**Conclusion**
This paper offers a well-rounded contribution to interpretable machine learning. It presents a simple, reproducible framework that bridges modern segmentation/detection tools with CBMs, demonstrating improved data efficiency and broader applicability. The authors were highly responsive to feedback, and the final set of experiments and clarifications resolve most of the initial concerns. Assuming the revisions outlined in the rebuttal are incorporated, I am supportive of accepting this paper.